# Molecular tuning boosts asymmetric C-C coupling for CO conversion to acetate

Jie Ding [1,9], Fuhua Li[1,9], Xinyi Ren[2], Yuhang Liu[3], Yifan Li [4], Zheng Shen[2], Tian Wang[5], Weijue Wang[2], Yang-Gang Wang [6], Yi Cui [4], Hongbin Yang [3] ✉, Tianyu Zhang[7] ✉ & Bin Liu [1,8] ✉

Electrochemical carbon dioxide/carbon monoxide reduction reaction offers a promising route to synthesize fuels and value-added chemicals, unfortunately their activities and selectivities remain unsatisfactory. Here, we present a general surface molecular tuning strategy by modifying $Cu_2O$ with a molecular pyridine-derivative. The surface modified $Cu_2O$ nanocubes by 4-mercaptopyridine display a high Faradaic efficiency of greater than 60% in electrochemical carbon monoxide reduction reaction to acetate with a current density as large as 380 mA/cm² in a liquid electrolyte flow cell. In-situ attenuated total reflectance surface-enhanced infrared absorption spectroscopy reveals stronger *CO signal with bridge configuration and stronger *OCCHO signal over modified $Cu_2O$ nanocubes by 4-mercaptopyridine than unmodified $Cu_2O$ nanocubes during electrochemical CO reduction. Density function theory calculations disclose that local molecular tuning can effectively regulate the electronic structure of copper catalyst, enhancing *CO and *CHO intermediates adsorption by the stabilization effect through hydrogen bonding, which can greatly promote asymmetric *CO-*CHO coupling in electrochemical carbon monoxide reduction reaction.

Acetate is an important chemical to manufacture food additives, solvents, medicine, etc[1,2]. The global demand for acetate was 17.3 million tons in 2019 and is predicted to reach 24.5 million tons by 2025[3]. Nowadays, methanol carbonylation still serves as the main method to produce acetate, which is energy-consuming and environmentally unfriendly[4–7]. On the other hand, electrochemical carbon dioxide/carbon monoxide reduction reaction ($CO_2RR/CORR$) powered by renewable electricity provides a greener approach to make acetate[8–12]. However, the selectivity and the partial current density towards the targeted acetate via $CO_2RR/CORR$ are still unsatisfactory, and the reaction pathway for $CO_2RR/CORR$ to acetate remains ambiguous.

Cu-based electrocatalysts have been intensively studied in $CO_2RR/CORR$ to make $C_{2+}$ fuels/chemicals, but they still suffer from poor selectivity[13]. Various strategies have been applied to tackle the poor selectivity challenge, including alloying, facet engineering, surface regulation, etc[14–23]. Among the different approaches, modifying Cu surface with certain small molecule was proposed as a promising route to modulate the reaction pathways for generation of $C_{2+}$ fuels/chemicals[24,25].

[1]Department of Materials Science and Engineering, City University of Hong Kong, Hong Kong SAR, China. [2]CAS Key Laboratory of Science and Technology on Applied Catalysis, Dalian Institute of Chemical Physics, Chinese Academy of Sciences, Dalian, China. [3]School of Materials Science and Engineering, Suzhou University of Science and Technology, Suzhou, China. [4]Vacuum Interconnected Nanotech Workstation, Suzhou Institute of Nano-Tech and Nano-Bionics, Chinese Academy of Sciences, Suzhou, China. [5]Department of Chemical & Biomolecular Engineering, National University of Singapore, Singapore, Singapore. [6]Department of Chemistry and Guangdong Provincial Key Laboratory of Catalysis, Southern University of Science and Technology, Shenzhen, China. [7]College of Environmental Science and Engineering, Beijing Forestry University, Beijing, China. [8]Department of Chemistry & Center of Super-Diamond and Advanced Films (COSDAF), City University of Hong Kong, Hong Kong SAR, China. [9]These authors contributed equally: Jie Ding, Fuhua Li. ✉e-mail: hbyang@dicp.ac.cn; tzhang@bjfu.edu.cn; bliu48@cityu.edu.hk

Molecules exhibited multiple-functions on $CO_2RR$ determined by the structure of molecules[26,27]. Molecular decoration is able to adjust the strength of intermediates adsorption and selectively stabilize certain intermediates through hydrogen bonding or enhance the adsorption of certain intermediates via confinement effect or regulating the catalyst's electronic structure to boost the catalytic performance[28–30]. However, insufficient understanding of the underlaying principle of small molecule modification on electrochemical $CO_2RR$/CORR remains a major factor hindering the development of this strategy.

In this work, we presented a universal strategy to modify Cu-based catalysts with 4-mercaptopyridine (pyS), including commercial Cu, $Cu_2O$, CuO as well as the as-prepared $Cu_2O$ nanocubes, which could dramatically enhance the catalytic performance for electrochemical CORR to produce acetate. The molecular tuning induced by 4-mercaptopyridine over Cu surface enhanced the adsorption of reaction intermediates in CORR, improved the hydrogenation of *CO, facilitated asymmetric *CO-*CHO coupling and thus stimulated the CO-to-acetate conversion. The surface modified $Cu_2O$ nanocubes by 4-mercaptopyridine ($Cu_2O$-pyS) exhibited a high acetate FE of >60% at a total current density of 380 mA cm$^{-1}$ in a liquid electrolyte flow cell, whose performance remained almost unchanged at 380 mA/cm$^2$ with an acetate FE over 60% for 100 h in a flow cell. The impressive CORR performance of $Cu_2O$-pyS originated from the boosted asymmetric C-C coupling induced by 4-mercaptopyridine modification, validated by in-situ ATR-SEIRAS measurements and DFT calculations.

## Results

### Catalyst preparation and CORR performance

$Cu_2O$ nanocubes were synthesized and further functionalized with 4-mercaptopyridine. $Cu_2O$ nanocubes with a mean size of ~50 nm was synthesized by a facile solution method and then modified with 4-mercaptopyridine under ultrasonication (Supplementary Fig. 1a). X-ray diffraction (XRD) of $Cu_2O$-pyS displays peaks at 29.6°, 36.4°, and 43.4°, which can be assigned to $Cu_2O$ (110), $Cu_2O$ (111) and $Cu_2O$ (200),

respectively (JCPDS file NO. 05-0667, Supplementary Fig. 1b). Compared to unmodified $Cu_2O$ nanocubes, the infrared spectrum of $Cu_2O$-pyS exhibits several characteristic peaks. The peaks at 788 cm$^{-1}$, 1460 cm$^{-1}$, and 1598 cm$^{-1}$ can be assigned to the stretching vibration of C=S, C=C and pyridine ring, respectively, demonstrating the successful introduction of 4-mercaptopyridine onto $Cu_2O$ nanocubes (Supplementary Fig. 1c). Scanning electron microscopy (SEM) and transmission electron microscopy (TEM) images (Supplementary Figs. 2a, b and 3a, b) indicate that the $Cu_2O$-pyS maintains the well-defined nanocube morphology after 4-mercaptopyridine modification. High-resolution energy-dispersive X-ray spectroscopy (EDS) line scan and mapping over $Cu_2O$-pyS show homogeneous distribution of Cu, O, N, and S elements, suggesting successful attachment of 4-mercaptopyridine on the surface of $Cu_2O$ nanocubes (Supplementary Figs. 2d and 3d). The clear lattice fringe of 0.213 nm observed on $Cu_2O$-pyS can be assigned to $Cu_2O$ (200) (Supplementary Figs. 2c and 3e). The electronic structure of $Cu_2O$-pyS was probed by X-ray photoelectron spectroscopy (XPS) and it was found that the binding energy of Cu in $Cu_2O$-pyS was shifted towards higher binding energies after 4-mercaptopyridine modification (Supplementary Fig. 4a). High-resolution N 1s and S 2p XPS spectra of $Cu_2O$-pyS display clear peaks originating from 4-mercaptopyridine. The high-resolution N 1s XPS spectrum can be deconvoluted into pyridine N (397.5 eV) and N with different degrees of H acceptance, respectively[31]. The high-resolution S 2p XPS spectrum displays a peak at 162 eV (Supplementary Fig. 4b), indicating the formation of Cu-S bond[32]. The presence of thiolate and the absence of metal-N bond corroborate that 4-mercaptopyridine is linked on $Cu_2O$ via the Cu-S bond.

The CORR was evaluated in a flow-cell electrolyzer to overcome the poor solubility of CO and the sluggish CO mass transport in aqueous solution (Supplementary Figs. 5 and 6)[33–35]. The electrochemical performance of $Cu_2O$-pyS was also evaluated. As presented in Fig. 1a & Supplementary Fig. 7, CO could be electrochemically reduced on $Cu_2O$-pyS to acetate, exhibiting a total current density of 380 mA/cm$^2$ with an acetate FE (FE$_{acetate}$) as high as 62% at −0.85 V versus reversible

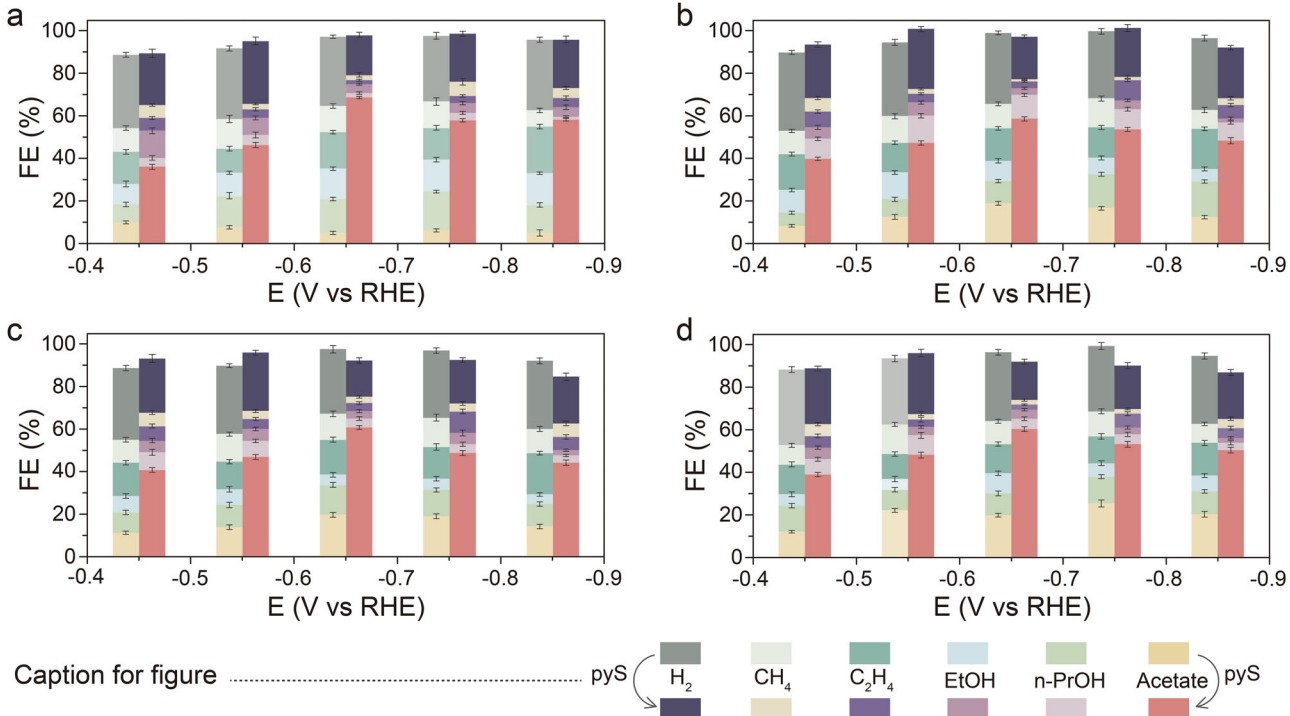

**Fig. 1 | Electrochemical CORR performance.** FE of various products for the as-prepared $Cu_2O$ nanocubes and $Cu_2O$-pyS (**a**), commercial Cu and c-Cu-pyS (**b**), commercial $Cu_2O$ and c-$Cu_2O$-pyS (**c**), and commercial CuO and c-CuO-pyS (**d**). Error bars represent the standard deviation of three replicate measurements.

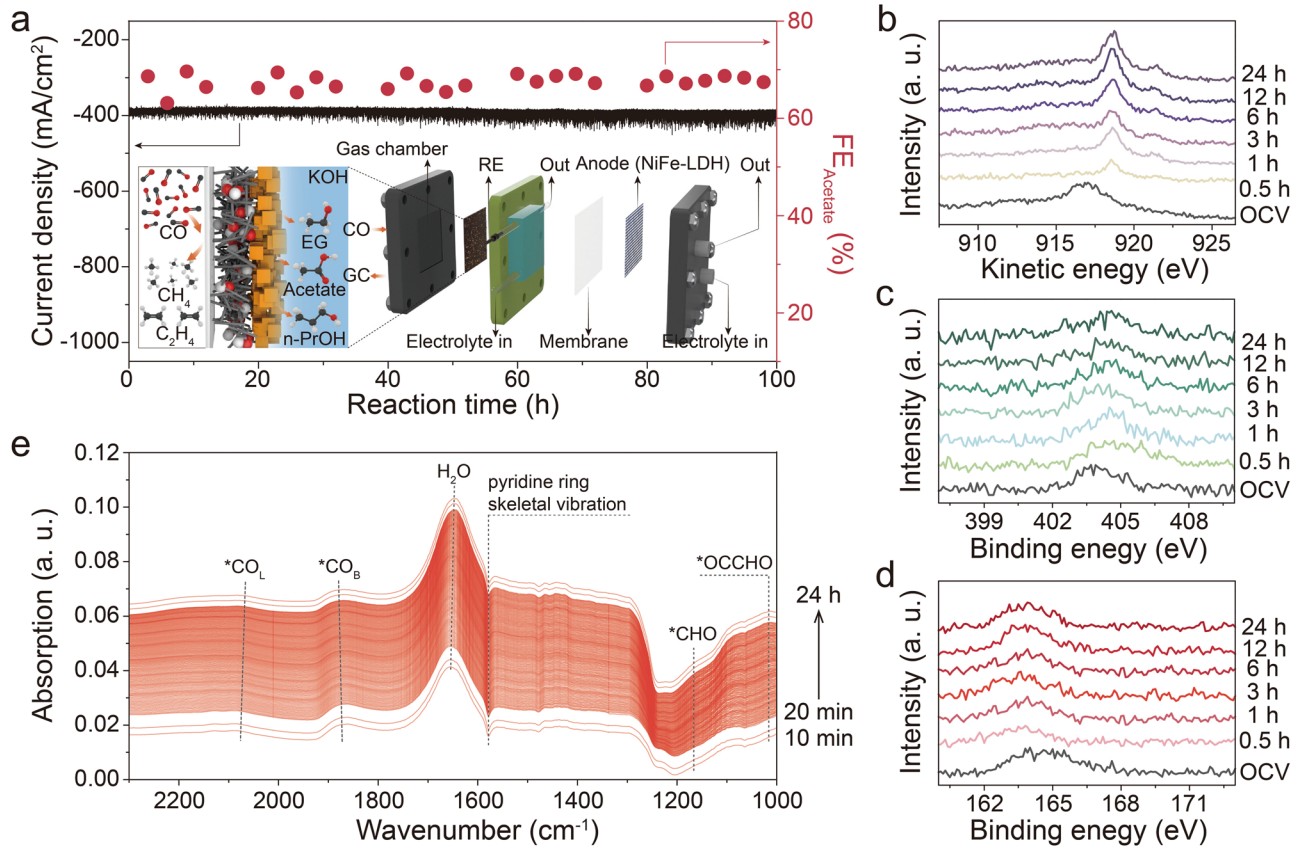

**Fig. 2 | Stability evaluation. a** CORR stability test of Cu$_2$O-pyS. Inset is the schematic illustration showing the flow cell. **b** Quasi in-situ time-dependent Cu LMM X-ray-excited Auger electron spectra of Cu$_2$O-pyS collected at −0.65 V vs. RHE. Quasi in-situ time-dependent high-resolution N 1s XPS spectra (**c**), and S 2p XPS spectra

(**d**) collected at −0.65 V vs. RHE. **e** In-situ ATR-SEIRAS spectra from 1000 cm$^{-1}$ to 2300 cm$^{-1}$ recorded over Cu$_2$O-pyS in CO-saturated 0.1 mol/L KOH solution at −0.65 V vs. RHE.

hydrogen electrode (RHE). A higher FE$_{acetate}$ of 70% could be realized at −0.65 V vs. RHE over Cu$_2$O-pyS. As compared to Cu$_2$O-pyS, the FE and partial current density towards acetate over unmodified Cu$_2$O nanocubes are much lower, indicating the critical role of 4-mercaptopyridine played in the CORR (the different electrochemically active surface area (ECSA) contribution could be excluded as shown in Supplementary Fig. 8). To verify the universality of the pyS-modification strategy, commercial Cu, Cu$_2$O, and CuO were also modified with 4-mercaptopyridine, denoted as c-Cu-pyS, c-Cu$_2$O-pyS, and c-CuO-pyS, respectively. After decoration of 4-mercaptopyridine, obvious enhancement in acetate production (selectivity and partial current density) could be observed for all commercial samples (Fig. 1b−d and Supplementary Fig. 9), confirming the generality of the pyS-modification approach. Besides activity and selectivity, stability is also crucial in CORR. The CORR stability of Cu$_2$O-pyS was evaluated in a 9 cm$^2$ flow cell. As displayed in Fig. 2a, the Cu$_2$O-pyS could be operated continuously at a current density of 380 mA/cm$^2$ and a FE$_{acetate}$ over 60% for 100 h with no noticeable activity decay, outperforming most of the reported state-of-the-art CORR electrocatalysts. It is worth mentioning that 2.08 mol of acetate, derived from superposition of current and Faradaic efficiency, could be produced in 100 h. SEM, TEM, and EDS measurements disclosed that the morphology and size of Cu$_2$O nanocubes and Cu$_2$O-pyS after CORR were changed (Supplementary Figs. 10 and 11). Both of Cu$_2$O nanocubes and Cu$_2$O-pyS were reduced to Cu under applied cathodic potentials during CORR (Supplementary Fig. 10a, b). Moreover, N and S could still be observed over Cu$_2$O-pyS after the CORR, indicating that 4-mercaptopyridine was stable under the applied cathodic potential (Supplementary Fig. 11c, d).

## Stability evaluation for CORR

To elucidate the structural evolution of Cu$_2$O-pyS during CORR, in-situ Raman spectroscopy was conducted. Raman peaks at 146 and 219 cm$^{-1}$ were observed over Cu$_2$O nanocubes and Cu$_2$O-pyS at the open circuit potential (OCP) (Supplementary Figs. 12 and 13), which are characteristic Raman peaks of Cu$_2$O[36]. With increase in the applied cathodic potential, the intensity of the Cu$_2$O characteristic Raman peaks decreased and finally disappeared, suggesting that Cu$_2$O was reduced to Cu. More importantly, the Raman peaks at 689 and 783 cm$^{-1}$, which could be assigned to Cu-S vibrational mode, resulting from 4-mercaptopyridine remained stable even at the applied potential of −1.0 V vs. RHE, which is already the CORR working potential for the catalyst, illustrating the excellent affinity of 4-mercaptopyridine over the copper surface. Additionally, Cu-C vibration (~300 cm$^{-1}$) could also be observed in the in-situ Raman spectra (as shown in Supplementary Fig. 13), resulting from the reactive intermediates in the CORR, which will be discussed later. Quasi in-situ Auger electron spectroscopy (AES) was performed to probe the valance state evolution of Cu during the reaction, where Cu$_2$O nanocubes was rapidly reduced to Cu$^0$ after 0.5 h (Fig. 2b). Similar results were also observed in commercial Cu$_2$O sample (Supplementary Fig. 14). Quasi in-situ time-dependent high-resolution N 1s and S 2p XPS spectra also confirmed that 4-mercaptopyridine could stay on the surface of Cu$_2$O catalyst at all times (Fig. 2c, d and Supplementary Fig. 15). To further elucidate the structural stability of Cu$_2$O-pyS, in-situ time-dependent ATR-SEIRAS measurement was carried out (Supplementary Fig. 16). DFT calculation was performed to identify the positions of the vibrational peaks (Supplementary Fig. 25). Compared to Cu$_2$O nanocube catalyst, a ATR-SEIRAS peak at around 1,600 cm$^{-1}$ could be observed on Cu$_2$O-pyS,

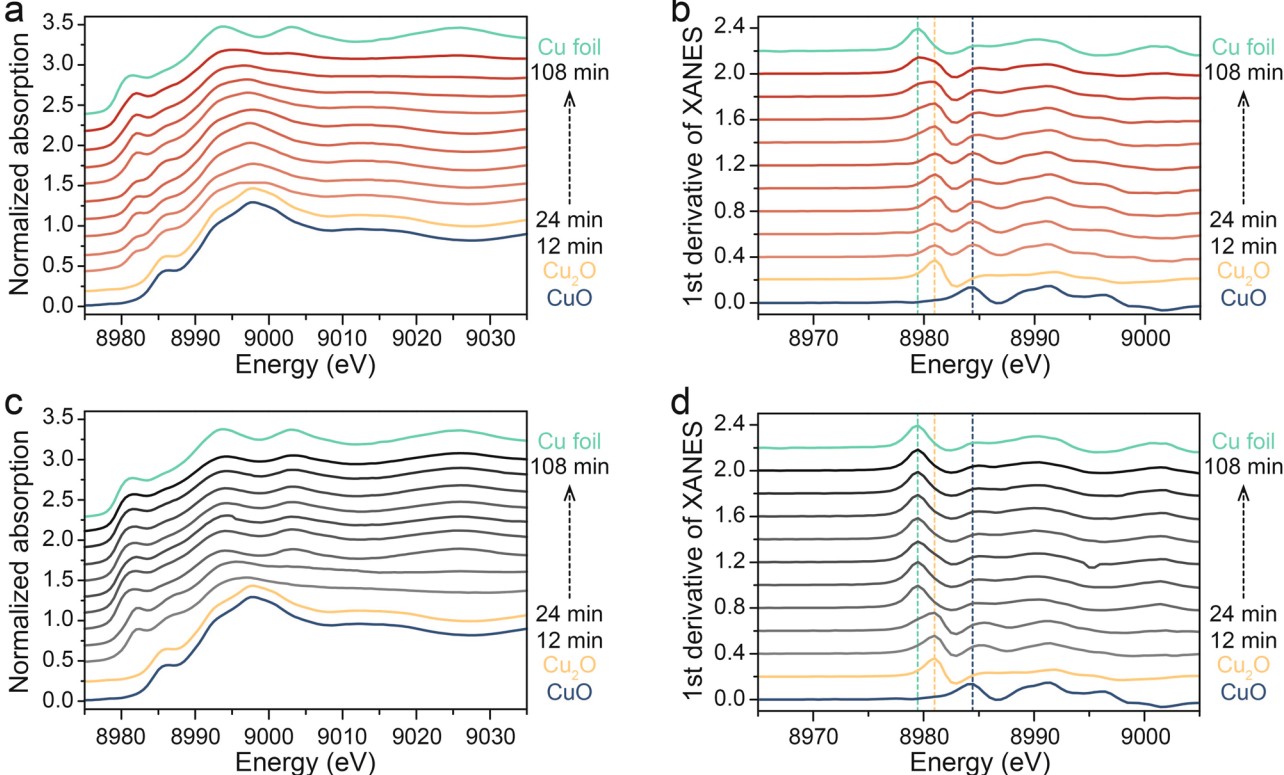

**Fig. 3 | In-situ Cu K-edge XANES measurements. a** In-situ Cu K-edge XANES spectra of Cu₂O nanocubes measured in CO-saturated KOH solution at −0.65 V vs. RHE. **b** The corresponding 1st derivative of XANES at −0.65 V vs. RHE. **c** In-situ Cu K-edge XANES spectra of Cu₂O-pyS measured in CO-saturated KOH solution at −0.65 V vs. RHE. **d** The corresponding 1st derivative of XANES at −0.65 V vs. RHE.

which can be assigned to pyridine ring skeletal vibration, revealing the successful attachment of 4-mercaptopyridine onto the Cu₂O surface. As shown in Fig. 2e, the pyridine ring skeletal vibration remained in 24 hours of CORR. Moreover, peaks for *CO$_L$, *CO$_B$, *CHO, *COCHO, and H₂O existed throughout the reaction, indicating that 4-mercaptopyridine was stable over the surface of the catalyst (Fig. 2e). X-ray absorption spectroscopy (XAS) was further conducted to disclose the structural and electronic evolution of Cu element. Figure 3a, c revealed that Cu₂O continued to be reduced in CORR. First derivative of X-ray absorption near edge structure (XANES) was analyzed to investigate the reduction of Cu₂O. The results showed that Cu₂O was converted into Cu under the applied cathodic potential, and the transformation was accelerated after decoration of 4-mercaptopyridine (Fig. 3b, d). The k³-weighted extended X-ray absorption fine structure (EXAFS) and the corresponding Fourier transform of EXAFS for Cu₂O nanocubes and Cu₂O-pyS recorded during CORR are shown in Supplementary Fig. 17. It is worth mentioning that Cu⁰ remained after its appearance, suggesting that Cu₂O (-pyS) derived Cu (-pyS) was the real catalyst for CORR to acetate.

## Mechanistic insights for CORR

To further shed light on the underlying role of pyS decoration on CORR, in-situ ATR-SEIRAS measurements under different applied cathodic potentials were performed to probe the reaction intermediates and the reaction pathway. Supplementary Figs. 18 and 19 show the ATR-SEIRAS spectra recorded over Cu₂O nanocubes. Compared to Cu₂O, the water peak at ~1625 cm⁻¹ is less pronounced on Cu₂O-pyS, suggesting that the presence of pyS can help to reduce water adsorption on the catalyst's surface, which may lower the undesired hydrogen evolution reaction (HER) over Cu₂O. When the applied cathodic potential increased to −0.2 V vs. RHE, several ATR-SEIRAS peaks started to appear over

Cu₂O-pyS. Two types of *CO peaks could be found in the ATR-SEIRAS spectra, one at 2080 cm⁻¹, which can be assigned to the linearly bound *CO (CO$_L$); the other at 1800 cm⁻¹, which can be ascribed to the bridge-bound *CO (Fig. 4a, b). Both *CO$_L$ and *CO$_B$ are reactive species in CORR towards acetate formation (Supplementary Figs. 20 and 21). The ratio of *CO$_B$/*CO$_L$ over Cu₂O-pyS is much larger than that over Cu₂O, implying that 4-mercaptopyridine could influence *CO adsorption configuration (Fig. 4c and Supplementary Figs. 20 and 21). Between the two types of *CO, *CO$_B$ is much easier to participate in the hydrogenation reaction due to its weak C=O bond, in contrast to the strong C≡O bond in *CO$_L$[37]. In addition, *CHO signal at 1,185 cm⁻¹ and *OCCHO signal at 1,040 cm⁻¹ could also be clearly identified over Cu₂O-pyS, which was further supported by the isotope labeling experiments (Supplementary Fig. 22). Previous reports have shown that *OCCHO plays as an important intermediate in CO₂RR/CORR to produce acetate[38]. Figure 4d shows that the ATR-SEIRAS peak ratio of *OCCHO/*CHO over Cu₂O-pyS is at least one order of magnitude larger than that over Cu₂O, justifying the boosted CORR to acetate over Cu₂O-pyS. The role of *OCCHO for CO-to-acetate conversion can be further confirmed by the observation that the *OCCHO coverage increased with applied cathodic potential. Two C-C coupling pathways were usually proposed in CORR to produce C₂₊ products (Fig. 4e): the asymmetric *CO-*CHO coupling for acetate formation and the symmetric *CO-*CO or *CHO-*CHO coupling for ethanol and ethylene formation. The detected *OCCHO intermediate can only exist in the pathway of asymmetric *CO-*CHO coupling, explaining the excellent activity and selectivity of CORR to acetate over Cu₂O-pyS. Similar phenomenon could also be observed in the case of commercial Cu₂O and the corresponding c-Cu₂O-pyS, highlighting the importance of 4-mercaptopyridine functionalization (Supplementary Fig. 26). Therefore, 4-mercaptopyridine decoration would be a general strategy

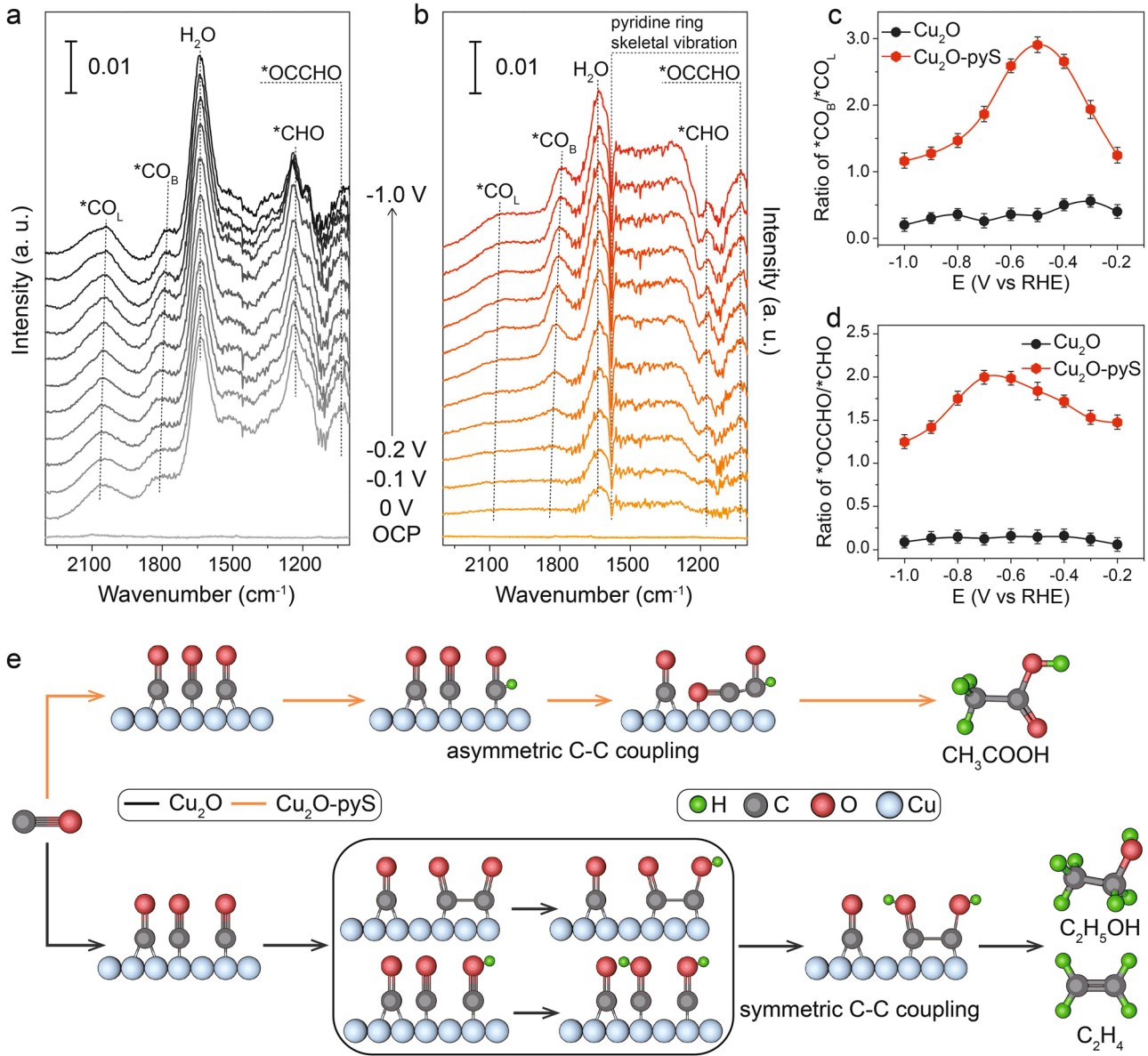

**Fig. 4 | In-situ ATR-SEIRAS measurements.** In-situ ATR-SEIRAS spectra recorded in a potential window from 0 to −1.0 V vs. RHE over $Cu_2O$ nanocubes (**a**) and $Cu_2O$-pyS (**b**) in 0.1 M KOH. **c** Calculated ratio of $^*CO_B/^*CO_L$ over $Cu_2O$ nanocubes and $Cu_2O$-pyS. **d** Calculated ratio of $^*OCCHO/^*CHO$ over $Cu_2O$ nanocubes and $Cu_2O$-pyS. **e** Schematic illustration showing the symmetric C-C coupling and asymmetric C-C coupling over $Cu_2O$ nanocubes and $Cu_2O$-pyS. Error bars in **c** and **d** represent s.d. for each data point ($n$ = 3 independent experiments), and points are average values.

to promote CO-to-acetate conversion via boosting asymmetric C-C coupling.

### Theoretical understanding

To further investigate the role of 4-mercaptopyridine played in CORR, the 4-mercaptopyridine was intentionally removed from the surface of $Cu_2O$ nanocubes by Ar plasma treatment (the successful removal of 4-mercaptopyridine on $Cu_2O$ nanocubes was confirmed by the disappearance of N and S signals in the XPS spectra as shown in Supplementary Fig. 23b, c). Note that the valence state of Cu decreased slightly after Ar plasma treatment (Supplementary Fig. 23a). Moreover, SEM and TEM images indicate that the morphology and size of $Cu_2O$ nanocubes were maintained after Ar plasma treatment (Supplementary Fig. 23d, e), while the CORR current density and acetate FE were found to decrease sharply after Ar plasma treatment, validating the important role played by 4-mercaptopyridine in CORR to produce

acetate (Supplementary Fig. 23f). On the other hand, thiophenol (phS) or 2,6-dimethylthiophenol (pyDMS) was also attached onto the surface of $Cu_2O$ nanocubes to explore the contribution of hydrophobicity, C, S or N on CORR (the modified sample is denoted as $Cu_2O$-phS and $Cu_2O$-pyDMS, Supplementary Fig. 24). The acetate FE of CORR over $Cu_2O$-phS significantly decreased, while that over $Cu_2O$-pyDMS remained nearly unchanged as compared to $Cu_2O$-pyS, suggesting that hydrophobicity itself did not play the most significant influence on the performance of CORR and the N atom in 4-mercaptopyridine rather than the S atom governed the CO-to-acetate conversion. To further exclude the contribution of steric effect, a molecule with similar functional group but longer chain (4-(Pyridin-4-yl)thiazole-2-thiol) was used to modify the surface of $Cu_2O$ nanocubes, the performance, as shown in Supplementary Fig. 24, was similar to that of $Cu_2O$-pyS and $Cu_2O$-pyDMS, suggesting that the role of steric effect was negligible. To gain a deeper insight into the reaction pathway as

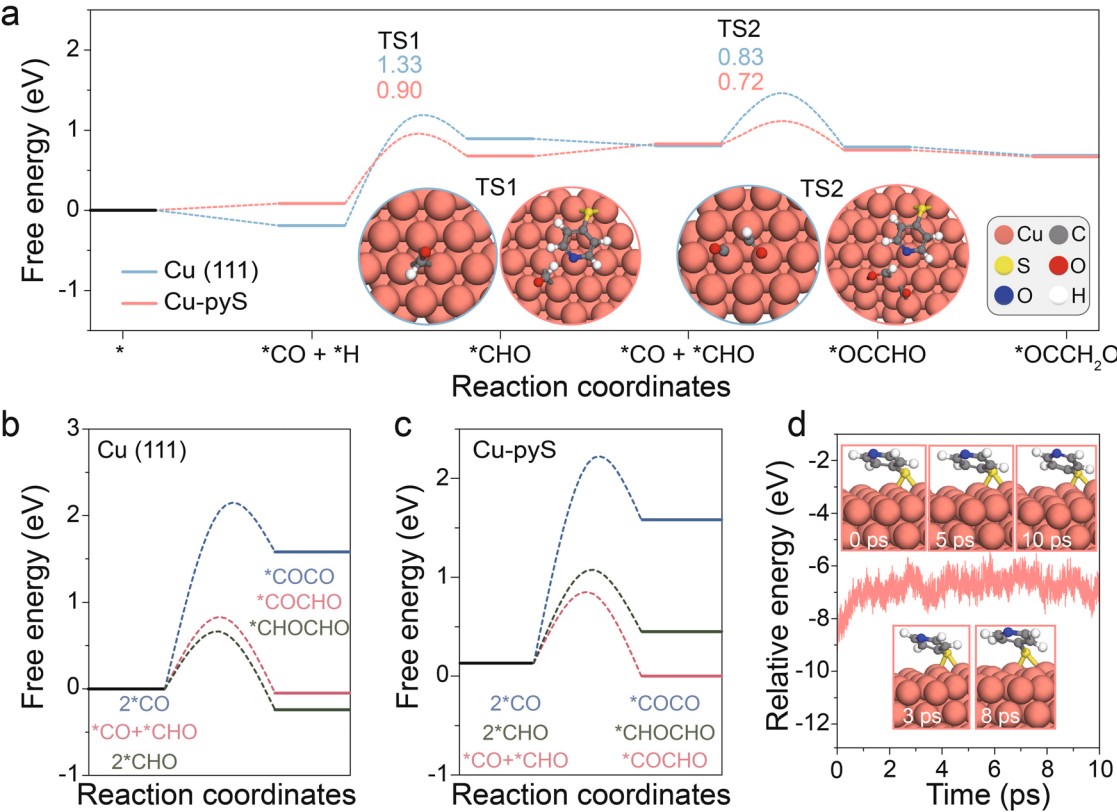

**Fig. 5 | DFT calculations. a** Free energy diagram showing the formation of *CHO and *OCCHO. Insets are the models of transitional states over Cu$_2$O and Cu$_2$O-pyS. **b** Free energy diagram of symmetric coupling and asymmetric coupling over Cu (111). **c** Free energy diagram of symmetric coupling and asymmetric coupling over Cu-pyS. **d** Ab initio molecular dynamics (AIMD) results of Cu$_2$O-pyS. Insets show the side view image of Cu$_2$O-pyS at 0 ps, 3 ps, 5 ps, 8 ps, and 10 ps.

well as the reaction mechanism, DFT calculations were carried out. Based on ex-situ XRD, in-situ XPS, in-situ Raman spectroscopy and in-situ X-ray absorption spectroscopy (XAS) characterizations, Cu (111) and 4-mercaptopyridine decorated Cu (111) surface were selected as the calculation models to represent Cu$_2$O nanocubes and Cu$_2$O-pyS, and the calculated free energy diagrams of CORR are displayed in Supplementary Fig. 27. It is obvious that the energy barrier of CO*-CHO* asymmetric coupling was much lower than that of CO*-CO* symmetric coupling over both Cu and Cu-pyS, suggesting that asymmetric coupling was conducive to acetate formation (Fig. 5b, c). Moreover, analyzing coverage of reaction intermediates (including *CO and *CHO) on C-C coupling process (*CO + *CO to *COCO and *CO + *CHO to *COCHO) indicates that the impact of *CHO coverage on ΔG and the corresponding activation barriers is stronger than that of *CO coverage in the *CO + *CHO to *COCHO process (Supplementary Fig. 28). Therefore, *CO hydrogenation to form *CHO is crucial. The rate-determining steps (RDSs) of CORR to form acetate over both Cu$_2$O and Cu$_2$O-pyS are *CO hydrogenation and *CO-*CHO coupling (Fig. 5a). Modifying Cu$_2$O with 4-mercaptopyridine can decrease the energy barrier of *CO hydrogenation and *CO-*CHO coupling from 1.33 to 0.9 eV and 1.16 to 0.81 eV, respectively. Note that starting from *CH$_2$CO, the generation of acetate is exothermic, while the production of ethanol and ethylene are endothermic, revealing the high selectivity towards acetate over Cu-pyS (Supplementary Fig. 27). To explore the origin of the accelerated hydrogenation reaction over Cu$_2$O-pyS, kinetic isotope effect (KIE) of H/D over Cu$_2$O nanocubes and Cu$_2$O-pyS were measured and compared. When H$_2$O was replaced by D$_2$O, the formation of acetate reduced obviously over Cu$_2$O nanocubes with a KIE value of 2.09 and 2.12 at −0.55 V vs. RHE and −0.65 V vs. RHE, respectively, suggesting that the activation and dissociation of water

were included in the RDS of CORR over Cu$_2$O. On the contrary, the acetate formation using D$_2$O remained nearly undisturbed over Cu$_2$O-pyS with a KIE value of 1.13 and 1.11 at −0.55 V vs. RHE and −0.65 V vs. RHE, respectively. The value approached 1, suggesting that the dissociation of water was not included in the RDS of CORR over Cu$_2$O-pyS and therefore the hydrogenation of *CO could be expedited over Cu$_2$O-pyS (Supplementary Fig. 29). DFT calculation disclosed that hydrogenation of *CO$_B$ was more energetically favorable than that of *CO$_L$, confirming that hydrogenation steps were accelerated over Cu$_2$O-pyS (Supplementary Fig. 31). Moreover, water dissociation was found to be propelled after the introduction of 4-mercaptopyridine (Supplementary Fig. 32a). N atom in 4-mercaptopyridine assisted to capture one H atom from water, facilitating water dissociation and the subsequent hydrogenation process (Supplementary Fig. 32b). Supplementary Fig. 30 shows charge redistribution occurred between the N atom in 4-mercaptopyridine and the H atom in *CHO and *OCCHO, indicating that 4-mercaptopyridine was beneficial to stabilize *CHO and *OCCHO. Supplementary Fig. 30 presents the differential charge density for various important CORR intermediates over Cu$_2$O and Cu$_2$O-pyS. *CO + *H was found more stable over Cu$_2$O, while *CHO and *OCCHO were more stable over Cu$_2$O-pyS. These observations suggest the role of hydrogen bond in stabilizing *CHO and *OCCHO, which can help to break the Brønsted-Evans-Polanyi (BEP) scaling relationship and boost CO-to-acetate conversion. The projected density of states (PDOS) analysis was then performed to get a better understanding of the reaction mechanism. For *CO + *H step, the Cu d-orbital of Cu$_2$O-pyS exhibits more overlapping with the H s-orbital, indicating favorable hydrogenation of *CO and undesirable HER over Cu$_2$O-pyS (Supplementary Fig. 33a, b). For *CHO, more overlapping between the Cu d-orbital of Cu$_2$O-pyS and the C p-orbital of *CHO was observed and

at the same time the N p-orbital of $Cu_2O$-pyS overlapped with the H s-orbital of *CHO, corroborating that *CHO was more stable over $Cu_2O$-pyS (Supplementary Fig. 33c, d). For *OCCHO, enhanced overlapping between the Cu d-orbital of $Cu_2O$-pyS and the C/O p-orbital of *CHO as well as the N p-orbital of $Cu_2O$-pyS and the H s-orbital of *CHO were noticed, justifying the stabilization effect of 4-mercaptopyridine on *OCCHO (Supplementary Fig. 33e, f). The enhanced adsorption to CORR intermediates over $Cu_2O$-pyS can be also supported by its higher d-band center (Supplementary Fig. 34). Based on the above investigations, the reaction pathways of CORR towards various products over $Cu_2O$ and $Cu_2O$-pyS are proposed as shown in Supplementary Figs. 35 and 36. To further examine the stability of $Cu_2O$-pyS, Ab initio molecular dynamics (AIMD) simulation was performed. No noticeable geometry change could be observed throughout the AIMD simulation, verifying the thermodynamic and kinetic durability of $Cu_2O$-pyS, matching well with the experiments (Fig. 5d).

## Discussion

To sum up, we have developed a general solution method to prepare 4-mercaptopyridine modified Cu-based catalysts, including commercial Cu, $Cu_2O$, CuO and the synthesized $Cu_2O$ nanocubes, which exhibit excellent electrochemical CORR performance to produce acetate. The total current density can reach approximately $380\ mA/cm^2$ with an acetate FE beyond 60% in a flow cell. In-situ ATR-SEIRAS observes stronger *CO signal with bridge configuration and stronger *OCCHO signal over $Cu_2O$-pyS than unmodified $Cu_2O$ during CORR. DFT calculations illustrate that local molecular modification can effectively tune the electronic structure of copper catalyst and strengthen *CO and *CHO intermediates adsorption by the stabilization effect through hydrogen bonding, which greatly promotes the asymmetric *CO-*CHO coupling in electrochemical CORR, resulting in an exceptional CO-to-acetate conversion performance.

## Methods

### Chemicals and materials

Copper (II) chloride ($CuCl_2$), sodium hydroxide (NaOH), ascorbic acid ($C_6H_8O_6$), 4-mercaptopyridine ($C_5H_5NS$), thiophenol ($C_6H_5SH$), 2,6-dimethylbenzenethiol ($C_8H_{10}S$) and isopropanol ($C_3H_8O$) were purchased from Sigma-Aldrich. All chemicals were used directly without further purification. De-ionized water (DI water) was obtained from Millipore Q water purification system.

### Synthesis of $Cu_2O$ nanocubes

$Cu_2O$ nanocubes were synthesized by an ascorbic acid reduction method at room temperature. Typically, 0.1 mmol of $CuCl_2$ was dissolved in 40 mL of DI water, followed by dropwisely adding 2.5 mL of NaOH aqueous solution (0.2 mol/L). Then the solution was stirred for 5 minutes, followed by dropwisely adding 2.5 mL of ascorbic acid solution (1 mol/L). The mixture was further stirred for another 5 minutes. Finally, the product was harvested by centrifugation and then washed with ethanol thoroughly.

### Synthesis of $Cu_2O$-pyS

The as-prepared $Cu_2O$ nanocubes were dispersed in N, N-dimethylformamide (DMF) and sonicated for 30 minutes at room temperature, followed by adding a solution containing 4-mercaptopyridine under inert atmosphere. Subsequently, the suspension was sonicated for another 3 hours. Finally, the product was harvested by centrifuge and then washed with DMF and ethanol thoroughly.

### Synthesis of c-Cu ($Cu_2O$/CuO)-pyS

The commercial Cu powder (25 nm, Sigma), $Cu_2O$ powder and CuO powder were dispersed in N, N-dimethylformamide (DMF) and sonicated for 30 minutes at room temperature, followed by adding a

solution containing 4-mercaptopyridine under inert atmosphere. Subsequently, the suspension was sonicated for another 3 hours. Finally, the product was harvested by centrifuge and then washed with DMF and ethanol thoroughly.

### Materials characterization

Powder XRD was performed on a Bruker D2 Phaser using Cu Kα radiation with a LYNXEYE detector at 30 kV and 10 mA. The morphological information was examined with field-emission SEM (FESEM, JEOL JSM-6700F). Sub angstrom-resolution high-angle annular dark field scanning transmission electron microscopy (HAADFSTEM) characterization was conducted on a JEOL JEMARM200F STEM with a guaranteed resolution of 0.08 nm. Raman spectra were recorded on a Renishaw INVIA Reflex Raman spectrometer using 514 nm laser as the excitation source. XPS measurements were carried out on a Thermofisher ESCALAB 250Xi photoelectron spectrometer (Thermofisher Scientific) using a monochromatic Al Kα X-ray beam (1,486.6 eV)[39].

### Electrode fabrication

A catalyst ink solution containing 10 mg of $Cu_2O$-pyS, 0.98 mL of DI water, 0.98 mL of isopropanol and 40 μl of Nafion ionomer solution (5 wt.% in $H_2O$) was mixed and sonicated for at least 3 hours. Then, the ink was deposited onto a carbon paper to achieve a catalyst loading of ~1 mg/$cm^2$. $Cu_2O$ electrode was also prepared using the same method. To ensure precise control of the catalytic layer thickness and achieve good uniformity, we employed ultrasonic spraying method to prepare the electrode. This involves two steps: (1) Ultrasonic dispersion to prepare catalyst paste; (2) Atomization and ultrasonic spraying of the catalyst slurry onto a support body, which can be either a gas diffusion layer or proton exchange membrane.

### Electrochemical measurements

CORR was performed in 1 M KOH solution in a three-channel flow cell. The anode and the cathode were separated by a hydroxide exchange membrane. Hg/HgO electrode was used as the reference electrode, while NiFeMoB developed by our research group[40] was deployed as the anode catalyst. The gas flow rate was 20 sccm. The products of CORR were quantified using a gas chromatography (Agilent 7890) equipped with a flame ionization detector (FID) and a thermal conductivity detector (TCD) and a high-performance liquid chromatography (Agilent Technologies 1260 Infinity) equipped with a RID detector. Acetate measurements were conducted only in the cathode chamber.

### In-situ ATR-SEIRAS measurements

The attenuated total reflectance surface enhanced infrared absorption spectroscopy (ATR-SEIRAS) measurements were performed on a Nicolet iS50 FTIR spectrometer equipped with a MCT detector cooled with liquid nitrogen and PIKE VeeMAX III variable angle ATR sampling accessory.

### Quasi in-situ XPS measurements

The X-ray photoelectron spectroscopy (XPS) measurements were performed on a SPECS NAP-XPS interconnected with a glovebox (Vigor Corp) (Supplementary Fig. 15). The CORR were performed in the glovebox, afterwards the sample was transferred to the XPS chamber without air exposure.

### In-situ Raman measurements

In-situ Raman spectroscopy measurements were conducted in a custom-designed three-electrode SERS flow cell with a saturated Ag/AgCl electrode as the reference electrode and a graphite rod as the counter electrode in the anode chamber. During the measurements, the electrolyte was constantly purged with CO gas and circulated across the cell using a peristaltic pump. The Raman measurements were performed on a LabRAM HR Evolution microscope (Horiba Jobin

Yvon) with a 532 nm laser, a 50× objective, a monochromator (600 grooves/mm grating), and a CCD detector[41].

## Computational methods

All spin-polarized DFT geometry optimizations were implemented under the description of generalized gradient approximation (GGA) based PBE-D3 functional with VASPsol model in the Vienna ab initio Simulation Package (VASP5.4.4). The $4 \times 4$ periodic rectangular supercell Cu (111) was built to support the 4-mercaptopyridine. The Brillouin zone was sampled by $3 \times 3 \times 1$ Monkhorst-Pack k-point scheme, and the cutoff was set as 450 eV. The convergence criteria were set as $10^{-5}$ eV in energy and 0.01 eV Å$^{-1}$ in force, respectively. A 20 Å vacuum space was added along the perpendicular direction to eliminate the effects of periodic images. The climbing image nudged elastic band (CI-NEB) method was employed with converged forces less than 0.05 eV/Å in VTST package to obtain transition states to derive the reaction free energy barriers[42]. During structure relaxation, the bottom two layers of copper atoms were immobilized, while all other ions were allowed to move freely to reasonably save computing resources. The canonical ensemble (NPT) ab initio molecular dynamic (AIMD) simulations were carried out in the Anderen thermostat at 298 K for 10 picoseconds (ps) with a time step of 1 femtosecond (fs).

In each of the electroreduction elementary steps, the reaction Gibbs free energy (ΔG) was calculated with consideration of thermal internal energy contribution according to:

$$\Delta G = \Delta E + \Delta E_{ZPE} - T\Delta S + G_U + G_{pH}$$

where $\Delta E$, $\Delta E_{ZPE}$ and $T\Delta S$ are the changes of electronic energy, the zero-point energy, and the temperature-entropy product in each elementary step. $\Delta G_U$ and $\Delta G_{pH}$ equaling to $-eU$ and $k_B T \ln 10 \times$ pH (here pH = 14) are the contribution of applied electrode potential and pH to ΔG. The entropy of H$^+$+e$^-$ pair is approximately half of H$_2$ entropy in standard condition.

## Data availability

All data are reported in the main text and supplementary materials. Source data are provided with this paper. All relevant data are available from the authors on reasonable request. Source data are provided with this paper.

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

## Acknowledgements

We acknowledge funding support from the City University of Kong Hong startup fund (9020003), ITF–RTH - Global STEM Professorship (9446006), the National Key Research and Development Program of China (No. 2022YFA1506200), CAS Project for Young Scientists in Basic Research (YSBR-022), the Strategic Priority Research Program of the Chinese Academy of Sciences (XDB36030200), the National Natural Science Foundation of China (Grant No. 22075195), the National Natural Science Foundation of China (No. 22208021, 21974103, 22102207, 2199152, and 21832004) and Photon Science Research center for Carbon Dioxide.

## Author contributions

J.D. and F.L. contributed equally. J.D., F.L., H.Y., T.Z. and B.L. conceived the project. B.L. supervised the project. J.D., X.R., Yu.L., Yi.L., Z.S., T.W., W.W., Y.W. and Y.C. performed the experimental study. T.Z., F.L. and Y.W. performed the theoretical study. J.D., F.L. H.Y., T.Z. and B.L. wrote the manuscript with support from all authors.

## Competing interests

The authors declare no competing interests.
