## [Peer Review File · Nature Communications]

Molecular Tuning Boosts Asymmetric C-C Coupling for CO Conversion to AcetateREVIEWER COMMENTS

Reviewer #1 (Remarks to the Author):

The authors claim Cu electrocatalysts decorated with mercaptopyrindines favor acetate formation at high rates (380 mA/cm² with FE>60).

The voltammetry, IR, TEM, Raman and x-ray work are well done and the story is generally believable. The DFT suggest the mercaptopyrindine stabilizes the CO and CHO intermediates, but I am skeptical of the argument that this favors acetate over ethylene or ethanol. It seems more likely that the oxides favor improved selectivity to alcohols or acetates, especially at high pH. Why no selectivity plots as a function of pH? What if hold the potential long enough to reduce any copper oxides before you start the reaction (or run cathodic current for several hours).

I am left with several concerns that should be addressed before recommending publication.

1. Aside from Figure 2a, there is no electrolyzer or system description. If acetate is only measured in the anolyte product, it seems likely that other products like ethylene or ethanol were formed at the cathode and were later oxidized to acetate at the anode. How do the authors know that acetate was formed directly at the cathode?
2. The authors state the the CuO and Cu₂O is concurrently reduced in the experiment. It seems like a challenge to focus on unstable oxides. It isn't clear if they have a role in acetate formation or if they simply provide a good precursor to make a low-coordinated Cu nanostructure? Also, it may be that the mercaptopyrindine adsorbs on oxides differently than Cu metal. This part isnt clear.
3. It could be that the hydrophobic nature of the mercapto pyridine is the main factor that promotes acetates. It could also be ascribed to CO COVERAGE and pH rather than the favorability of bridge bound CO and formation of *OCCHO due to the presence of mercaptopyrindine. The authors should consider the literature that describes acetate formation without adsorbed ligands.
4. Figure 1 is impossible for me to read. This should be simplified with colors that are clearly different for different products. For example. is hard to tell the difference between C₂H₄ and n-propanol.

There are several other english and grammatical errors that detract from the manuscript. It is also strange that the authors call the RHE the "reversible" hydrogen electrode. "RHE" is usually used to designate "relative" meaning that it is adjusted for pH.

Reviewer #2 (Remarks to the Author):

In this work, Ding and co-workers presented a surface molecular tuning strategy by modifying Cu₂O

nanocubes with pyS, which significantly improved the selectivity of acetate in electrochemical CORR. Authors modified several Cu catalysts with pyS and all samples showed higher acetate selectivity and partial current density, implying the generality of pyS-modification strategy. Using in-situ XPS, XANES and Raman, authors investigated the characteristics of pyS-modified catalysts and confirmed the stability of pyS under the reaction condition. Besides, authors employed ATR-SEIRAS to demonstrate that the higher ratio of COB was beneficial to *CO-*CHO coupling, promoting the formation of acetate rather than ethylene and ethanol. Overall, the reactivity results are interesting, though not necessarily the best among current literature. The mechanistic discussion is more speculative in nature (see comments below), and thus the reviewer doesn't consider this work rise to the standards of this journal. More detailed comments:

1. Ln132-136, the authors claimed that 2.08 mol acetate could be produced in 100 h. Was the produced acetate separated and collected to verify this claim?
2. What is the morphology of Cu₂O nanocubes during or after reaction? Since Cu₂O is certainly reduced during the reaction, the morphology of the Cu₂O is less relevant than reduced catalyst. In Fig. S8b, the authors marked Cu(111) and Cu(OH)₂(311), rather than Cu(100), which is expected of Cu nanocubes.
3. Authors claimed that *COB was much easier to be hydrogenated (Line 207). In the previous work (ACS Catal., 2018, 8, 7507-7516), however, *COB was considered as an unreactive species in CORR. How do authors explain the contradiction between such a view and the claim in this work? The cited ref.37 about HCOOH oxidation on Pt does not seem to support authors' point very well. Thus, stronger evidence should be provided.
4. Why isn't S-Cu vibrational mode observed in the Raman spectra of Fig. S10 and 11? How are the 4-mercaptopyridine bands observed compared with the molecular spectrum (this should be provided in the SI to aid the peak assignment)?
5. Peak assignments in the IR spectra are quite speculative. First of all, why is the pyridine ring skeletal mode negative? The features assigned as *CHO and *OCCHO are weak and broad, how can the authors be sure of these assignments? Neither of the bands attributed to *CHO and *OCCHO appear to shift with potential in Figure 4a and b, suggesting that these are not surface bond species. Additional experimental evidence is needed to support the attribution of correlative signals in SEIRA spectra (Figure 4a and 4b), such as isotope labeling. If the observation of *CHO and *OCCHO doesn't stand on solid ground, the mechanistic analysis based on DFT calculations appears shaky.
6. Multiple literature reports suggest again bridge-bonded CO on Cu is not active in the CORR, e.g., ACS Catal., 2018, 8, 7507-7516. The authors argue the opposite, so strong evidence needs to be provided to support this claim.
7. If *COB is indeed the key intermediate, why is the ratio of *COB/*COL higher at the pyS-modified surface and how does pyS affect CO adsorption configuration?

Response to Reviewers

We are very grateful to the critical comments and constructive suggestions provided by the reviewers, which shall significantly help to improve the quality of our work. Our manuscript has been revised accordingly, and the changes are highlighted by yellow colour in the text. The following list our responses to the comments and suggestions from the reviewers. All the original comments are given in blue colour (italic) and our responses in black colour.

Reviewer #1 (Remarks to the Author):

The authors claim Cu electrocatalysts decorated with mercaptopyridines favor acetate formation at high rates (380 mA/cm² with FE>60). The voltammetry, IR, TEM, Raman and x-ray work are well done and the story is generally believable. The DFT suggest the mercaptopyridine stabilizes the CO and CHO intermediates, but I am skeptical of the argument that this favors acetate over ethylene or ethanol. It seems more likely that the oxides favor improved selectivity to alcohols or acetates, especially at high pH. Why no selectivity plots as a function of pH? What if hold the potential long enough to reduce any copper oxides before you start the reaction (or run cathodic current for several hours).

Response: We sincerely appreciate the reviewer for the time and efforts spent in assessing our work. To fully address the reviewer's concerns, detailed responses and supporting data are provided below.

1. The DFT suggest the mercaptopyridine stabilizes the CO and CHO intermediates, but I am skeptical of the argument that this favors acetate over ethylene or ethanol. It seems more likely that the oxides favor improved selectivity to alcohols or acetates, especially at high pH.

Response: We thank the reviewer for the valuable suggestion. Structure re-construction of CuO and Cu₂O are important for understanding the CORR/CO₂RR over Cu based catalysts. It has suggested that low-coordinated Cu sites can be generated over CuO and Cu₂O under applied cathodic potentials, which exhibit superior CORR/CO₂RR

performance compared to metallic Cu. In this work, a series of *in-situ* measurements have been performed to understand the structure reconstruction process. As indicated by time-dependent high-resolution XPS, XAS and ATR-SEIRAS data, over both Cu₂O and Cu₂O-pyS samples, structure reconstruction occurred, both of which changed from Cu₂O to oxide-derived Cu. The oxide-derived Cu is the real catalyst for CORR to acetate. The acetate formation process over oxide-derived Cu in our work (**Figure R1-1**) follows the same pathway as that over other reported Cu based catalysts not modified with 4-mercaptopyridine (*Angew. Chem. Int. Ed.*, 2022, 61, e202111167; *ACS Catal.*, 2022, 12, 5275; *Energy Fuels*, 2023, 37, 7904). **Figure R1-1** describes the asymmetric *CO-*CHO coupling for acetate formation and the symmetric *CO-*CO or *CHO-*CHO coupling for ethanol and ethylene formation. The *OCCHO intermediate can only exist in the pathway of asymmetric *CO-*CHO coupling, which was clearly detectable over Cu₂O-pyS, explaining the excellent activity and selectivity of CORR to acetate over Cu₂O-pyS. Similar phenomenon could also be observed in the case of commercial Cu₂O and the corresponding c-Cu₂O-pyS, highlighting the importance of 4-mercaptopyridine functionalization to promote CORR to form acetate (**Figure R1-2**). Many systematic studies have been conducted to investigate the influence of Cu surface state on selectivity of C₂₊ products during electrochemical CO₂RR. The research findings indicate that, regardless of the initial states, all copper oxide/hydroxide based materials are reduced to Cu⁰ during CO₂RR. Moreover, the starting copper oxide/hydroxide based materials undergo fragmentation into irregular nanosized Cu under applied cathodic potentials. Such fragmentation resulting from an oxidation-reduction cycle greatly facilitates C–C coupling and thus contributes to enhanced C₂₊ products selectivity (*J. Am. Chem. Soc.*, 2020, 142, 4213; *ACS Cent. Sci.*, 2019, 5, 12, 1998). The selectivity of C₂₊ products is not directly influenced by pH, as indicated by numerous studies. Koper et al. observed that cations did not influence proton reduction at low overpotentials; however, acidic cations could undergo hydrolysis under alkaline surface pH conditions, leading to a second regime of proton reduction. The activity and onset potential of water reduction reaction are correlated with acidity of cations (*J. Am. Chem. Soc.*, 2022, 144, 1589). Furthermore, DFT calculations were employed to

propose a mechanism for electrochemical reduction of CO on Cu(100) that aligned well with the experimental observations, demonstrating independence of pH on the formation of C₂₊ products in CORR (*Angew. Chem. Int. Ed.*, 2013, 52, 1). Additionally, we also performed CORR in different electrolytes (0.1 M, 1.0 M, and 5.0 M KOH) to explore the effect of pH on CORR, and the results are displayed in **Figure R1-3**. Increasing pH of electrolyte lowered the undesired hydrogen evolution reaction (from 20% to 10%), and the C₂₊ products increased from 77% to 80%. The FE of acetate exhibited the highest 70% in 0.1 M KOH. Thus, pH of electrolyte only slightly influenced CORR, which is consistent with the early study (*Angew. Chem. Int. Ed.*, 2020, 59, 4464-4469).

Figure R1-1. Schematic illustration showing the symmetric C-C coupling and asymmetric C-C coupling over Cu₂O nanocubes and Cu₂O-pyS.

Figure R1-2. (a) *In-situ* ATR-SEIRAS spectra recorded over commercial Cu₂O in CO-saturated 0.1 M KOH solution at different applied cathodic potentials. (b) *In-situ* ATR-

SEIRAS spectra recorded over c-Cu₂O-pyS in CO-saturated 0.1 M KOH solution at different applied cathodic potentials.

Figure R1-3. FE of various CORR products over the as-prepared Cu₂O-pyS in different pH of KOH electrolyte at -0.65 V vs. RHE.

2. Why no selectivity plots as a function of pH?

Response: The reviewer's constructive comment is greatly appreciated. Accordingly, we performed CORR in different electrolytes (0.1 M, 1.0 M, and 5.0 M KOH) to explore the effect of pH on CORR, and the results are displayed in **Figure R1-4**. Increasing pH of electrolyte lowered the undesired hydrogen evolution reaction (from 20% to 10%), and the C₂₊ products increased from 77% to 80%. The FE of acetate exhibited the highest 70% in 0.1 M KOH. Thus, pH of electrolyte only slightly influenced CORR, which is consistent with Xu's work (*Angew. Chem. Int. Ed.*, 2020, 59, 4464-4469).

Figure R1-4. FE of various CORR products over the as-prepared Cu₂O-pyS in different pH of KOH electrolyte at -0.65 V vs. RHE.

What if hold the potential long enough to reduce any copper oxides before you start the reaction (or run cathodic current for several hours).

Response: We thank the reviewer for the valuable suggestion. If we held the cathodic potential long enough time to reduce copper oxide before starting the reaction, the CORR performance maintained more or less the same. To verify the chemical status of Cu₂O and Cu₂O-pyS catalysts following long-term CORR, we performed *in-situ* X-ray absorption spectroscopy (XAS). As shown in **Figure R1-5**, first derivative of X-ray absorption near edge structure (XANES) was analyzed to investigate the reduction of Cu₂O. The results showed that Cu₂O was converted into Cu under the applied cathodic potential, and the transformation was accelerated after decoration of 4-mercaptopyridine (**Figure R1-5 b, d**). The k^3 -weighted extended X-ray absorption fine structure (EXAFS) and the corresponding Fourier transform of EXAFS for Cu₂O nanocubes and Cu₂O-pyS recorded during CORR are displayed in **Figure R1-6**. It is worth mentioning that Cu⁰ remained after its appearance, suggesting that Cu₂O (-pyS) derived Cu (-pyS) was the real catalyst for CORR to acetate. *Quasi in-situ* Auger electron spectroscopy (AES) was performed to probe the valence state evolution of Cu during the reaction, where Cu₂O nanocubes were rapidly reduced to Cu⁰ after 0.5 hour

(Figure R1-7a), which was maintained during the 24 hours of CORR, consistent with the findings of the *in situ* XAS experiment. Moreover, to further elucidate the structural stability of Cu₂O-pyS, *in-situ* time-dependent ATR-SEIRAS measurement was carried out (Figure R1-7b).

Figure R1-5. (a) *In-situ* Cu K-edge XANES spectra of Cu₂O nanocubes measured in CO-saturated KOH solution at -0.65 V vs. RHE. (b) The corresponding 1st derivative of XANES at -0.65 V vs. RHE. (c) *In-situ* Cu K-edge XANES spectra of Cu₂O-pyS measured in CO-saturated KOH solution at -0.65 V vs. RHE. (d) The corresponding 1st derivative of XANES at -0.65 V vs. RHE.

Figure R1-6. The k^3 -weighted EXAFS in K -space for Cu₂O nanocubes (a) and Cu₂O-pyS (b) measured in CO-saturated KOH solution at -0.65 V vs. RHE.

Figure R1-7. (a) *Quasi in-situ* time-dependent Cu LMM X-ray-excited Auger electron spectra of Cu₂O-pyS collected at -0.65 V vs. RHE. *Quasi in-situ* time-dependent high-resolution N 1s XPS spectra, (b) *In-situ* ATR-SEIRAS spectra from 1000 cm⁻¹ to 2300 cm⁻¹ recorded over Cu₂O-pyS in CO-saturated 0.1 M KOH solution at -0.65 V vs. RHE.

I am left with several concerns that should be addressed before recommending publication.

1. Aside from Figure 2a, there is no electrolyzer or system description. If acetate is only measured in the anolyte product, it seems likely that other products like ethylene or ethanol were formed at the cathode and were later oxidized to acetate at the anode. How do the authors know that acetate was formed directly at the cathode?

Response: We thank the reviewer for raising the critical comment. In the revised manuscript, the image of CO reduction electrolyzer was provided. As shown in **Figure R1-8**, CORR was performed in a three-channel flow cell using 1 M KOH as the electrolyte. The anode and cathode chambers were separated by a proton exchange membrane (Nafion[®] 117) to avoid crossover between reaction products at the anode and the cathode. Therefore, we only measured acetate in the cathodic chamber.

Figure R1-8. The photograph showing the CO reduction electrolyzer.

2. The authors state the CuO and Cu₂O is concurrently reduced in the experiment. It seems like a challenge to focus on unstable oxides. It isn't clear if they have a role in acetate formation or if they simply provide a good precursor to make a low-coordinated Cu nanostructure? Also, it may be that the 4-mercaptopyridine adsorbs on oxides differently than Cu metal. This part isn't clear.

Response: We thank the reviewer for the valuable comments. Structure re-construction of CuO and Cu₂O are important for understanding the CORR/CO₂RR over Cu based catalysts. It has suggested that low-coordinated Cu sites can be generated over CuO and Cu₂O under applied cathodic potentials, which exhibit superior CORR/CO₂RR performance compared to metallic Cu. In this work, a series of *in-situ* measurements have been performed to understand the structure reconstruction process. As indicated by time-dependent high-resolution XPS, XAS and ATR-SEIRAS data, over both Cu₂O and Cu₂O-pyS samples, structure reconstruction occurred, both of which changed from Cu₂O to oxide-derived Cu. The oxide-derived Cu is the real catalyst for CORR to acetate. The main difference between the two samples is the time to archiving stable catalyst. There might be some differences in the adsorption configuration of 4-mercaptopyridine on Cu oxides and Cu, but based on *in-situ* ATR-SEIRAS measurements, we observed that the wavenumber (1580 cm⁻¹), which related to 4-mercaptopyridine adsorption, did not show obvious shift in the process of Cu oxides

reduction to oxide-derived Cu under cathodic bias, suggesting similar adsorption configuration of 4-mercaptopyridine on both Cu oxides and oxide-derived Cu via metal-sulfur bond (*Analytical Sciences*, 23, 2007, 787; *Journal of Molecular Structure*, 2011, 991, 103; *Electrochimica Acta*, 2013, 114, 7; *Surface Science*, 1999, 430, 206). Besides, *in-situ* Raman spectroscopy and *quasi in-situ* time-dependent high-resolution XPS measurements also confirmed that 4-mercaptopyridine was stable under the applied cathodic potential throughout the CORR.

*3. It could be that the hydrophobic nature of the 4-mercaptopyridine is the main factor that promotes acetates. It could also be ascribed to CO COVERAGE and pH rather than the favorability of bridge bound CO and formation of *OCCHO due to the presence of 4-mercaptopyridine. The authors should consider the literature that describes acetate formation without adsorbed ligands.*

Response: We thank the reviewer for the valuable comments and suggestion. 4-Mercaptopyridine modification could lower the undesired hydrogen evolution reaction (HER) and promote CORR to acetate. To understand the impact of hydrophobic nature of 4-mercaptopyridine on CORR, a series of molecules like thiophenol (phS) and 2,6-dimethylthiophenol (pyDMS) were also attached onto Cu₂O nanocubes to explore the contribution of hydrophobicity towards CORR. The contact angles are 118.3 °, 60.8 °, 60.3 °, 73.6 ° and 72.7 ° over Cu₂O nanocubes, Cu₂O-phS, Cu₂O-pyS, Cu₂O-pyDMS and Cu₂O-dmBTH, respectively. As shown in **Figure R1-9**, compared to Cu₂O, the acetate FE of CORR over Cu₂O-phS significantly decreased, while that over Cu₂O-pyDMS and Cu₂O-dmBTH remained nearly unchanged as compared to Cu₂O-pyS, which suggested that hydrophobicity itself did not play the most significant influence on the performance of CORR. To understand the coverage of reaction intermediates (including *CO and *CHO) on the C-C coupling process (*CO + *CO to *COCO and *CO + *CHO to *COCHO), DFT calculations were conducted. **Figure R1-10** shows the free energy of *CO + *CO to *COCO (**Figure R1-10a**) and *CO + *CHO to *COCHO (**Figure R1-10b**) for different coverages of *CO and *CHO. As shown in **Figure R1-10**, with increasing *CO and *CHO coverages, both the free energy and the

corresponding activation barriers of transition states decrease. It is worthy to note that the impact of *CHO coverage on ΔG and the corresponding activation barriers is stronger than that of *CO coverage in the *CO + *CHO to *COCHO process. Importantly, with increasing *CO coverage, *CO prefers to adsorb on the top site of Cu other than the hollow site. Accordingly, we performed CORR in different electrolytes (0.1 M, 1.0 M, and 5.0 M KOH) to explore the effect of pH on CORR, and the results are displayed in **Figure R1-11**. Increasing pH of electrolyte lowered the undesired hydrogen evolution reaction (from 20% to 10%), and the C₂₊ products increased from 77% to 80%. The FE of acetate exhibited the highest 70% in 0.1 M KOH. Thus, pH of electrolyte only slightly influenced CORR, which is consistent with Xu's work (*Angew. Chem. Int. Ed.*, 2020, 59, 4464-4469). As indicated by time-dependent high-resolution XPS, XAS and ATR-SEIRAS data, over both Cu₂O and Cu₂O-pyS samples, structure reconstruction occurred, both of which changed from Cu₂O to oxide-derived Cu. The oxide-derived Cu is the real catalyst for CORR to acetate. The acetate formation process over oxide-derived Cu in our work (**Figure R1-12**) follows the same pathway as that over other reported Cu based catalysts not modified with 4-mercaptopyridine (*Angew. Chem. Int. Ed.*, 2022, 61, e202111167; *ACS Catal.*, 2022, 12, 5275; *Energy Fuels* 2023, 37, 7904). **Figure R1-12** describes the asymmetric *CO-*CHO coupling for acetate formation and the symmetric *CO-*CO or *CHO-*CHO coupling for ethanol and ethylene formation. The *OCCHO intermediate can only exist in the pathway of asymmetric *CO-*CHO coupling, which was clearly detectable over Cu₂O-pyS, explaining the excellent activity and selectivity of CORR to acetate over Cu₂O-pyS. Similar phenomenon could also be observed in the case of commercial Cu₂O and the corresponding c-Cu₂O-pyS, highlighting the importance of 4-mercaptopyridine functionalization to promote CORR to form acetate (**Figure R1-13**).

Figure R1-9. Different molecules used to modify the surface of Cu_2O , and the corresponding CORR performance.

Figure R1-10. Theoretical insights into the catalytic mechanism. a, b, Calculated Gibbs free energy diagrams and the corresponding energy barrier of transition states for $^*\text{CO}$ - $^*\text{CO}$ coupling to $^*\text{COCO}$ and $^*\text{CO}$ - $^*\text{CHO}$ coupling to $^*\text{COCHO}$ with various coverage of $^*\text{CO}$ and $^*\text{CHO}$ over Cu surface.

Figure R1-11. FE of various products over the as-prepared Cu₂O-pyS in different pH of KOH electrolyte at -0.65 V vs. RHE.

Figure R1-12. Schematic illustration showing the symmetric C-C coupling and asymmetric C-C coupling over Cu₂O nanocubes and Cu₂O-pyS.

Figure R1-13. (a) *In-situ* ATR-SEIRAS spectra recorded over commercial Cu_2O in CO-saturated 0.1 M KOH solution at different applied cathodic potentials. (b) *In-situ* ATR-SEIRAS spectra recorded over *c*- Cu_2O -pyS in CO-saturated 0.1 M KOH solution at different applied cathodic potentials.

4. Figure 1 is impossible for me to read. This should be simplified with colors that are clearly different for different products. For example, is hard to tell the difference between C_2H_4 and *n*-propanol.

Response: We thank the reviewer for the valuable suggestion. We apologize that we used the camouflage colors for different products. According to the reviewer's suggestion, we have used distinct colors to describe different products (**Figure R1-14**).

Figure R1-14. FE of various products for the as-prepared Cu₂O nanocubes and Cu₂O-pyS (d), commercial Cu and c-Cu-pyS (b), commercial Cu₂O and c-Cu₂O-pyS (c), and commercial CuO and c-CuO-pyS (d), and. Error bars represent the standard deviation of 3 replicate measurements.

5. There are several other English and grammatical errors that detract from the manuscript. It is also strange that the authors call the RHE the "reversible" hydrogen electrode. "RHE" is usually used to designate "relative" meaning that it is adjusted for pH.

Response: We thank the reviewer for the critical comments. According to which, we have carefully checked the entire manuscript and supplementary information and corrected the English and grammatical errors. RHE is reversible hydrogen electrode potential (*Pure Appl. Chem.*, 2014, 86, 259). It is defined as the potential of platinum in the solution of any pH value, not limiting to the H ion concentration of 1 (pH = 0), and the partial pressure of hydrogen is 1 atmosphere. RHE electrode potential is widely used in present literatures.

Reviewer #2 (Remarks to the Author):

*In this work, Ding and co-workers presented a surface molecular tuning strategy by modifying Cu₂O nanocubes with pyS, which significantly improved the selectivity of acetate in electrochemical CORR. Authors modified several Cu catalysts with pyS and all samples showed higher acetate selectivity and partial current density, implying the generality of pyS-modification strategy. Using in-situ XPS, XANES and Raman, authors investigated the characteristics of pyS-modified catalysts and confirmed the stability of pyS under the reaction condition. Besides, authors employed ATR-SEIRAS to demonstrate that the higher ratio of CO_B was beneficial to *CO-*CHO coupling, promoting the formation of acetate rather than ethylene and ethanol. Overall, the reactivity results are interesting, though not necessarily the best among current literature. The mechanistic discussion is more speculative in nature (see comments below), and thus the reviewer doesn't consider this work rise to the standards of this journal. More detailed comments:*

Response: We sincerely appreciate the reviewer for the time and efforts spent in assessing our work. We have conducted a comprehensive study on the comments and suggestions raised by this reviewer. To fully address the concerns of the reviewer, we have provided detailed responses and supporting data as shown below.

1. Ln132-136, the authors claimed that 2.08 mol acetate could be produced in 100 h. Was the produced acetate separated and collected to verify this claim?

Response: We thank the reviewer for raising the question. The calculation of 2.08 mol of acetate that could be produced in 100 h is derived from the superposition of current and Faradaic efficiency, rather than directly isolating the product. To enhance the rigor of the paper, we have revised this statement.

2. What is the morphology of Cu₂O nanocubes during or after reaction? Since Cu₂O is certainly reduced during the reaction, the morphology of the Cu₂O is less relevant than reduced catalyst. In Fig. S8b, the authors marked Cu(111) and Cu(OH)₂(311), rather than Cu(100), which is expected of Cu nanocubes.

Response: We express our gratitude to the reviewer for raising the question. The Cu₂O nanocubes undergo a significant reconstruction process during CORR and are transformed into oxide-derived Cu. **Figure R2-1** demonstrates that the morphology of Cu₂O nanocubes after CORR exhibits fragmented Cu, no longer existing as Cu nanocubes. Therefore, in Fig S8b, we have labelled Cu(111) and Cu(OH)₂(311), instead of Cu(100).

Figure R2-1. The morphology of Cu₂O nanocubes after CORR.

*3. Authors claimed that *CO_B was much easier to be hydrogenated (Line 207). In the previous work (ACS Catal., 2018, 8, 7507-7516), however, *CO_B was considered as an unreactive species in CORR. How do authors explain the contradiction between such a view and the claim in this work? The cited ref.³⁷ about HCOOH oxidation on Pt does not seem to support authors' point very well. Thus, stronger evidence should be provided.*

Response: We express our gratitude to the reviewer for the insightful comments. To date, extensive research has been conducted to evaluate various C-C coupling pathways and elucidate the reaction mechanisms involved in the formation of C₂ products (*Proc. Natl. Acad. Sci.*, 2022, 119, e2202931119; *Adv. Sens. Energy Mater.*, 2022, 1, 100023; *Nano Energy*, 2023, 111, 108404; *ACS Catal.*, 2016, 6,219). These pathways include

the *CO-*CO pathway (*Nat. Catal.*, 2022, 5, 564; *J. Phys. Chem. Lett.*, 2016, 7, 1471; *Proc. Natl. Acad. Sci.*, 2017, 114, 1795; *Nat. Commun.*, 2019, 10, 32), *CO-*COH pathway (*J. Am. Chem. Soc.*, 2016, 138, 483), and *CO-*CHO pathway (*ACS Catal.*, 2018, 8, 1490). Furthermore, other studies have proposed alternative reaction mechanisms such as the *CO-CHOH pathway (*Nat. Commun.*, 2019, 10, 32), *CH₂-*CH₂ pathway (*Angew. Chem. Int. Ed.*, 2013, 125, 2519), and *COH-*COH pathway (*ACS Omega*, 2021, 6, 17839; *Nat. Catal.*, 2020, 3, 478). Characterizing the adsorption configuration of CO on catalyst's surface and comprehending its role in electrochemical reaction are crucial for understanding the C-C coupling reaction in CORR/CO₂RR. More specifically, the role of *CO_B has sparked a heated debate. Previous study has demonstrated through *in-situ* Raman spectroscopy that *CO_B acts as an inert species on Cu surface and does not significantly contribute to hydrocarbon and C₂₊ oxygenate formation (*ACS Catal.*, 2018, 8, 7507). However, in terms of the role played by different adsorption modes of CO in C-C coupling, numerous studies have also demonstrated that bridge CO_B on Cu surface can facilitate *CO hydrogenation and *CHO formation. Consequently, an energetically favorable pathway to acetate is established through asymmetric C-C coupling between *CO_L and *CHO (*Nature*, 2020, 577, 509; *Energy Fuels*, 2023, 37, 7904). This mechanism is supported by the computationally predicted shift of *CO adsorption from the top-site configuration on Cu (or Cu-rich) surface to the bridge sites of Cu-Pd bimetallic surface, which is also correlated with a reduction in the CO hydrogenation barrier (*ACS Catal.*, 2022, 12, 5275). In addition, to further confirm whether CO is involved in C-C coupling to acetate, the intensities of CO_B and CO_L were monitored using *in-situ* ATR-SEIRAS, and the results are displayed in **Figure R2-2**. The *in-situ* ATR-SEIRAS spectra were acquired during continuous CO purging, with spectra being continuously recorded until the CO peaks reaching a virtually constant state based on Xu's method (*Catal. Sci. Technol.*, 2021, 11, 6825). While maintaining the potential at -0.65 V_{RHE}, the purge gas was switched from CO to Ar. At the moment of switching, we initiated collection of the SEIRAS spectrum. **Figure 2-2a** shows the time-dependent SEIRAS spectra. The intensities of the *CO_{atop} and *CO_{bridge} peaks are normalized with respect to their

maximum values as shown in **Figure 2-2b**. The results show that both the $*CO_{atop}$ and $*CO_{bridge}$ species are consumed under Ar atmosphere at a bias of $-0.65 V_{RHE}$, suggesting that both $*CO_{atop}$ and $*CO_{bridge}$ are reactive species in CORR towards acetate formation. According to the molecular orbital theory, the highest occupied molecular orbital (HOMO) of CO is the almost non-bonding σ orbital that is localized to C, and the lowest unoccupied molecular orbital (LUMO) is the antibonding π orbital. The involvement in formation of σ and π orbital are C and O atoms. **Figure R2-3a** displays the molecular orbital energy level of CO molecule. The 1σ orbital is primarily localized on O, thus, it is essentially non-bonding or weakly bonding orbital. 2σ orbital is bonding orbital. The 1π orbital is a double degenerate pair of π bonding orbitals, which mainly has the property of $C2p$ orbital. The HOMO of CO is a 3σ orbital, which has the property of $C2p_z$ orbital, basically a non-bonding orbital, and is localized to C. The LUMO is a double degenerate pair of antibonding π orbitals, which has the property of $C2p$ orbital (**Figure R2-3b**). This combination of frontier orbitals (a pair of empty π orbitals that are essentially localized to carbon's fully filled σ orbitals) is the reason to form M-CO. CO is a ligand with a variety of coordination modes (**Figure R2-3c**), usually involving bridging one, two, or three metal atoms, and the expansion frequency of CO follows the following sequence: $MCO > M_2CO > M_3CO$. The coordination information shows that CO has excellent adsorption on the surface of catalyst. From previous studies, the CO adsorption energy and the C-O vibration frequency are related to the adsorption sites. Moreover, the specific wavenumber of CO, adsorbed on catalyst's surface, depends heavily on the adsorption energy. The stronger the adsorption is, the weaker the C=O vibration, the easier to participate in the hydrogenation reaction. Besides, we apologize that we made a mistake of using ref. 37 to support our point and the corresponding reference has been updated in the revised manuscript.

Figure R2-2. (a) *In-situ* ATR-SEIRAS spectra on Cu₂O-pyS as a function of time. (b) Temporal evolution of normalized peak area of *CO_{atop} and *CO_{bridge}.

Figure R2-3. (a & b) Orbital interactions of CO molecule. (c) Adsorption modes of CO on metal surface. (d) The peak position assignments of different CO adsorption modes.

4. Why isn't S-Cu vibrational mode observed in the Raman spectra of Fig. S10 and 11? How are the 4-mercaptopyridine bands observed compared with the molecular spectrum (this should be provided in the SI to aid the peak assignment)?

Response: We thank the reviewer for the critical comment and suggestion. In fact, Cu-S vibrational mode (around ~146 and 269 cm⁻¹) can be seen in the Raman spectrum of Cu₂O-pyS. We apologize that we did not assign the weak S-Cu vibrational mode in the Raman spectra. The FTIR spectrum of 4-mercaptopyridine has been provided in Figure S1c to aid the peak assignment (**Figure R2-5**).

Figure R2-4. Raman spectra of (a) Cu₂O nanocubes and (b) Cu₂O-pyS recorded in Ar-saturated 1 M KOH solution at different applied cathodic potentials.

Figure R2-5. FTIR spectra of the as-synthesized Cu₂O nanocubes and Cu₂O-pyS.

*5. Peak assignments in the IR spectra are quite speculative. First of all, why is the pyridine ring skeletal mode negative? The features assigned as *CHO and *OCCHO are weak and broad, how can the authors be sure of these assignments? Neither of the bands attributed to *CHO and *OCCHO appear to shift with potential in Figure 4a and b, suggesting that these are not surface bond species. Additional experimental evidence is needed to support the attribution of correlative signals in SEIRA spectra (Figure 4a and 4b), such as isotope labeling. If the observation of *CHO and *OCCHO doesn't stand on solid ground, the mechanistic analysis based on DFT calculations appears shaky.*

Response: We thank the reviewer for the critical comments. Previous works have indicated that the inverted IR peaks can be attributed to the content of adsorbed species lower than its background (*J. Am. Chem. Soc.*, 2022, 144, 6613; *Chem*, 2021,7, 1297; *Angew. Chem. Int. Ed.*, 2022, 61, e202206233). Under the applied potential of structure reconstruction, Cu₂O was transformed to oxide-derived Cu. In the initiation of this structure reconstruction, some 4-mercatopyridine molecules would drop from the catalyst's surface. Thus, compared to fresh 4-mercatopyridine modified Cu₂O, the pyridine ring content on oxide-derived Cu-pyS would be lower, and the pyridine ring skeletal mode displayed a negative peak in ATR-SEIRAS spectra (**Figure R2-6**). It is worthy to note that the bonded 4-mercatopyridine on oxide-derived Cu-pyS is stable during CORR, which is supported by *quasi in-situ* time-dependent high-resolution XPS and *in-situ* ATR-SEIRAS measurements (**Figure R2-7**). According to our DFT calculation results (**Figure R2-8**), the positions of the vibrational peaks of *CHO and *COCHO are located at about 1185 cm⁻¹ and 1050 cm⁻¹, respectively, which match well with our FTIR results. Additionally, the peak assignment of *CHO and *COCHO can also be supported by previous works (*Nat. Commun.*, 2023, 14, 7681; *J. Am. Chem. Soc.*, 2022, 144, 547). The peaks of *in-situ* ATR-SEIRAS can be influenced by various factors, including the Stark effect (*Nat. Commun.*, 2022, 13, 2656), CO coverage (*Angew. Chem.*, 2022, 134, e202111167), and isotope mass effect (*Angew. Chem. Int. Ed.*, 2022, 61, e202207197). In our work, the positions of the vibrational peaks of *CHO and *COCHO did not show obvious shift with the applied potential on the electrodes (oxide-derived Cu and oxide-derived Cu-pyS); similar results were observed in previous work (*Nat. Commun.*, 2022, 13, 3754). The possible reason is that the coverage of *CHO and *COCHO on the electrode surface is insufficient within the potential range of *in-situ* ATR-SEIRAS, thereby hindering the occurrence of interactions between adsorbents. Furthermore, D₂O was used in *operando* ATR-SEIRAS measurements to aid the peak assignment. As shown in **Figure R2-9**, when D₂O was used instead of H₂O, the position of *CHO and *COCOH moved by 25 cm⁻¹ and 21 cm⁻¹ towards lower wavenumbers due to isotopic redshift caused by the mass

effect. Thus, it is confirmed that the peak at $\sim 1185\text{ cm}^{-1}$ and $\sim 1019\text{ cm}^{-1}$ can be ascribed to the $\ast\text{CHO}$ and $\ast\text{COCO}$ intermediates in CORR.

Figure R2-6. *In-situ* ATR-SEIRAS spectra from 1000 cm^{-1} to 2300 cm^{-1} recorded over $\text{Cu}_2\text{O-pyS}$ in CO-saturated 0.1 M KOH solution at -0.65 V vs. RHE.

Figure R2-7. (a) *Quasi in-situ* time-dependent Cu LMM X-ray-excited Auger electron spectra of $\text{Cu}_2\text{O-pyS}$ collected at -0.65 V vs. RHE. *Quasi in-situ* time-dependent high-resolution N $1s$ XPS spectra (b) and S $2p$ XPS spectra (c) collected at -0.65 V vs. RHE.

Figure R2-8. Simulated IR peaks of important intermediates. (a) *CO simulated peak over top site of Cu (111). (b) *CO simulated peak over hollow site of Cu (111). (c) *CHO simulated peak over Cu (111). (d) *COCHO simulated peak over Cu (111). (e) *CO simulated peak over top site of Cu-pyS. (f) *CO simulated peak over hollow site of Cu-pyS. (g) *CHO simulated peak over Cu-pyS. (h) *COCHO simulated peak over Cu-pyS.

Figure R2-9. In-situ ATR-SEIRAS spectra recorded in a potential window from -0.1 to -0.9 V vs. RHE over Cu₂O-pyS (b) in 0.1 M KOD.

6. Multiple literature reports suggest again bridge-bonded CO on Cu is not active in the CORR, e.g., *ACS Catal.*, 2018, 8, 7507-7516. The authors argue the opposite, so strong evidence needs to be provided to support this claim.

Response: We thank the reviewer for the critical comment. Previous work (*ACS Catal.*, 2018, 8, 7507) suggested that $*CO_B$ was inert species on Cu surface. However, we also note a different interpretation regarding the correlation between C_{2+} product formation and $*CO_B$ on Cu surface in CORR (*Nature*, 2020, 577, 509; *Nat. Catal.*; 2022, 5, 251) and CO_2RR (*Sci. Bull.*, 2021, 66, 62), which suggest that an appropriate linear-to-bridge ratio of surface CO^* may benefit C-C coupling. Besides, *in-situ* ATR-SEIRAS measurements were performed to support that $*CO_B$ indeed played a significant role in CORR, and the results are displayed in **Figure R2-10**. The *in-situ* ATR-SEIRAS measurements were performed at $-0.65 V_{RHE}$. During continuous CO purging, ATR-SEIRAS spectra were collected until the CO peaks reached virtually constant. While maintaining the potential at $-0.65 V_{RHE}$, the purge gas was switched from CO to Ar. At the point of switching, we started to collect the spectrum as shown in **Figure R2-10a**. The intensities of the $*CO_{atop}$ and $*CO_{bridge}$ peaks are normalized with respect to their maximum values (**Figure R2-10b**). The results show that the $*CO_{atop}$ and $*CO_{bridge}$ are consumed under Ar atmosphere, suggesting that both $*CO_{atop}$ and $*CO_{bridge}$ are reactive species in CORR.

Figure R2-10. (a) *In-situ* ATR-SEIRAS spectra on Cu₂O-pyS as a function of time.

7. If $*CO_B$ is indeed the key intermediate, why is the ratio of $*CO_B/*CO_L$ higher at the pyS-modified surface and how does pyS affect CO adsorption configuration?

Response: We thank the reviewer for the critical comment. The adsorption behavior of CO on a catalyst's surface plays a pivotal role in various chemical reactions. In this context, it has been observed that bridge adsorption of CO generally exhibits greater strength compared to linear adsorption (*Nat. Commun.*, 2022, 13, 2656). This implies that at a low coverage of CO on catalyst's surface, CO tends to form bridge structure with neighboring atoms or molecules. However, as the coverage of CO increases, it transforms towards linear adsorption where individual CO molecules bind directly to metal sites. Interestingly, 4-mercatopyridine modification on the surface of Cu can significantly impact the adsorption properties of CO. The inclusion of 4-mercatopyridine leads to an accelerated consumption of CO, which reduces its overall coverage on the catalyst's surface. Consequently, any remaining adsorbed CO species exhibit a higher inclination towards bridge adsorption.

REVIEWER COMMENTS

Reviewer #2 (Remarks to the Author):

The reviewer appreciates the authors' effort to address the comments raised. However, there are still a few questions need to be addressed before the manuscript becomes publishable.

1. In the response to the 3rd question from reviewer 2 and as shown in Figure R2-2(b), the authors claim that the bridge-site *COB is an active species due to the decrease of its peak area after the purge gas was switched from CO to Ar. The reason given by the authors to such phenomena is that the surface *CO species were consumed by electrochemical reduction reactions, thus justifying the reactivity of *COB. It is unclear to the reviewer whether the Ar purge gas was continuously kept on during the collection of the SEIRAS spectrums in Figure R2-2(b), if so, the possibility of the *CO's desorption caused by the Ar purge (rather than consumption) leading to the decrease of its peak should be considered. A control experiment at a relatively high potential where CORR reactions do not occur could be conducted to further justify the authors' arguments. Also, the peak areas of *COL was also observed to decrease in the Ar atmosphere. As both *COL and *COB peak areas decrease, the reviewer finds it unconvincing to exclude *COL as a reactive species.

2. In the response to the 7th question of reviewer 2, the authors state that the higher *COB/*COL ratio on pyS-modified surface is due to the bridged-sited *CO as a more preferred configuration at lower *CO coverages, which is a consequence of an accelerated consumption rate. A different explanation may also be put forward that *COL is, in fact, the active species and as *COL is consumed at a higher rate on the modified surface, its surface concentration is lower and thus a higher *COB/*COL ratio is observed.

3. From Ln 246-254, the authors demonstrate that the pyridine structure is of crucial significance in improving the catalysts' performance, it seems from the calculations the N atom plays an important role. The reviewer would like the authors to comment on whether N atoms of different electronic structures (e.g. N from pyrrole or amides) or other elements possessing H-bond forming abilities would show similar effects.

4. The reviewer thanks the authors for the revision on Figure 1. As a minor question, there are some mistakes in the notes of the figures. For example, the reviewer believes that in the note of Figure 1 the "as-prepared Cu₂O nanocubes and Cu₂O-pyS" should be annotated as "(a)", rather than "(d)", as shown below.

Reviewer #3 (Remarks to the Author):

In my opinion, the authors have addressed all concerns from the reviewer 1. Overall, the results presented are interesting, from both mechanistic and performance perspectives. I have some minor suggestions about the experimental details so other groups can reproduce the data.

- The authors used a carbon gas diffusion layer that can operate for 100 hours without flooding problem.

This is quite surprising. Details about this material should be provided.

- The authors mentioned that the flow cell is 9 cm². Is this the area of the catalyst (electrode) or the size of the cell? Details about the flow cell configuration should be provided. From figure 2, it looks like there is no gas chamber or gas flow channel.

- How did the authors collect liquid products (only in catholyte or in both catholyte and anolyte)?

- How did the author deposit catalyst ink onto the gas diffusion layer?

Response to Reviewers

We are very grateful to the critical comments and constructive suggestions provided by the reviewers, which shall significantly help to improve the quality of our work. Our manuscript has been revised accordingly, and the changes are highlighted by yellow colour in the text. The following list our responses to the comments and suggestions from the reviewers. All the original comments are given in blue colour (italic) and our responses in black colour.

Reviewer #2 (Remarks to the Author):

The reviewer appreciates the authors' effort to address the comments raised. However, there are still a few questions need to be addressed before the manuscript becomes publishable.

Response: We sincerely appreciate the reviewer for the time and efforts spent in further assessing our work. To fully address the reviewer's concerns, detailed responses and supporting data are provided below.

*1. In the response to the 3rd question from reviewer 2 and as shown in Figure R2-2(b), the authors claim that the bridge-site $*CO_B$ is an active species due to the decrease of its peak area after the purge gas was switched from CO to Ar. The reason given by the authors to such phenomena is that the surface $*CO$ species were consumed by electrochemical reduction reactions, thus justifying the reactivity of $*CO_B$. It is unclear to the reviewer whether the Ar purge gas was continuously kept on during the collection of the SEIRAS spectrums in Figure R2-2(b), if so, the possibility of the $*CO$'s desorption caused by the Ar purge (rather than consumption) leading to the decrease of its peak should be considered. A control experiment at a relatively high potential where CORR reactions do not occur could be conducted to further justify the authors' arguments. Also, the peak areas of $*CO_L$ was also observed to decrease in the Ar atmosphere. As both $*CO_L$ and $*CO_B$ peak areas decrease, the reviewer finds it unconvincing to exclude $*CO_L$ as a reactive species.*

Response: We thank the reviewer for the helpful comments. To clarify the reviewer's

concerns, *in-situ* ATR-SEIRAS spectra were acquired during continuous CO purging, until the CO peaks reached a virtually constant state. Upon reaching the maximum CO peak, the purge gas was switched from CO to Ar, and the applied cathodic potential was stopped. **Figure R2-1a** shows the time-dependent ATR-SEIRAS spectra. The intensities of the $*\text{CO}_L$ and $*\text{CO}_B$ peaks are normalized with respect to their maximum values as shown in **Figure R2-1b**. The results showed that Ar purge played a negligible influence to the decrease of CO peaks. Moreover, once a potential of -0.65 V vs. RHE was applied, both the $*\text{CO}_L$ and $*\text{CO}_B$ species were consumed under Ar atmosphere, indicating that both $*\text{CO}_L$ and $*\text{CO}_B$ were reactive species in CORR towards acetate formation (**Figure R2-2**).

Figure R2-1. (a) *In-situ* ATR-SEIRAS spectra on $\text{Cu}_2\text{O-pyS}$ as a function of time. (b) Temporal evolution of normalized peak area of $*\text{CO}_L$ and $*\text{CO}_B$.

Figure R2-2. (a) *In-situ* ATR-SEIRAS spectra on $\text{Cu}_2\text{O-pyS}$ as a function of time. (b) Temporal evolution of normalized peak area of $*\text{CO}_L$ and $*\text{CO}_B$.

2. In the response to the 7th question of reviewer 2, the authors state that the higher $*CO_B/*CO_L$ ratio on pyS-modified surface is due to the bridged-sited $*CO$ as a more preferred configuration at lower $*CO$ coverages, which is a consequence of an accelerated consumption rate. A different explanation may also be put forward that $*CO_L$ is, in fact, the active species and as $*CO_L$ is consumed at a higher rate on the modified surface, its surface concentration is lower and thus a higher $*CO_B/*CO_L$ ratio is observed.

Response: We thank the reviewer for the constructive comment. We performed *in-situ* ATR-SEIRAS measurements to probe the adsorption behavior of CO on catalyst's surface. As shown in **Figure R2-3** and **Figure R2-4**, the consumption rate of $*CO_L$ and $*CO_B$ were similar under identical experimental conditions (ATR-SEIRAS spectra were collected after switching CO to Ar at -0.65 V vs. RHE). This suggested that the higher $*CO_B/*CO_L$ ratio should not result from a higher $*CO_L$ consumption rate on pyS-modified Cu_2O .

Figure R2-3. (a) *In-situ* ATR-SEIRAS spectra on Cu_2O -pyS as a function of time. (b) Temporal evolution of normalized peak area of $*CO_L$ and $*CO_B$.

Figure R2-4. (a) *In-situ* ATR-SEIRAS spectra on Cu₂O-pyS as a function of time. (b) Temporal evolution of normalized peak area of *CO_L and *CO_B.

3. From Ln 246-254, the authors demonstrate that the pyridine structure is of crucial significance in improving the catalysts' performance, it seems from the calculations the N atom plays an important role. The reviewer would like the authors to comment on whether N atoms of different electronic structures (e.g. N from pyrrole or amides) or other elements possessing H-bond forming abilities would show similar effects.

Response: We thank the reviewer for the constructive comments and suggestions. The primary objective of our original experimental design was to substantiate the underlying origin behind the observed performance variation in Cu-based catalysts that had been subjected to modification with small molecules. Thiophenol (phS) or 2,6-dimethylthiophenol (pyDMS) was also attached onto the surface of Cu₂O nanocubes to explore the contribution of hydrophobicity, C, S or N on CORR (the modified sample is denoted as Cu₂O-phS and Cu₂O-pyDMS, **Figure R2-5**). The acetate FE of CORR over Cu₂O-phS significantly decreased, while that over Cu₂O-pyDMS remained nearly unchanged as compared to Cu₂O-pyS, suggesting that hydrophobicity itself did not play the most significant influence on the performance of CORR and the N atom in 4-mercaptopyridine rather than the S atom governed the CO-to-acetate conversion. To further exclude the contribution of steric effect, a molecule with similar functional group but longer chain (4-(Pyridin-4-yl)thiazole-2-thiol) was used to modify the surface of Cu₂O nanocubes, the performance, as shown in **Figure R2-5**, was similar to that of

Cu₂O-pyS and Cu₂O-pyDMS, suggesting that the role of steric effect was negligible. The reviewer posed an insightful inquiry regarding the specific type of N in a small molecule that exerted a dominant influence on the electroreduction of CO to acetate. To answer the reviewer's question, we performed additional theoretical calculations. As depicted in **Figure R2-6**, when different types of N are adsorbed onto the surface of Cu₂O, there shows a significant alteration in the density of states and the D band center. Consequently, utilizing different small molecules for catalyst modification may engender variations in catalytic performance. We further explored how different N species adsorbed on Cu-based catalysts impacted CO hydrogenation to *CHO (**Figure R2-7**). The calculation results revealed that disparate types of N species exhibited different abilities to promote or inhibit CO hydrogenation. Lastly, we also investigated how different N species adsorbed on Cu surface affected H₂O hydrolysis (**Figure R2-8**). The findings showed that different types of N species adsorbed on Cu surface could either facilitate or impede hydrolytic dissociation. The reviewer's valuable comments and suggestions are greatly appreciated.

Figure R2-5. Different molecules used to modify the surface of Cu₂O. The corresponding contact angle (top) and CORR performance (down).

Figure R2-6. D-band center of Cu₂O (a), Cu₂O-pyS (pyridine N) (b), Cu₂O-pyrrole (c) and Cu₂O-amides N (d), respectively.

Figure R2-7. (a) Free energy diagram of bridge *CO hydrogenation over Cu, Cu₂O-pyS (pyridine N), Cu₂O-pyrrole, and Cu₂O-amides N. (b) The transition states of the hydrogenation step.

Figure R2-8. (a) Free energy diagram of water dissociation over Cu, Cu₂O-pyS (pyridine N), Cu₂O-pyrrole, and Cu₂O-amides N. (b) The side view images of water dissociation process over Cu, Cu₂O-pyS (pyridine N), Cu₂O-pyrrole, and Cu₂O-amides N.

4. The reviewer thanks the authors for the revision on Figure 1. As a minor question, there are some mistakes in the notes of the figures. For example, the reviewer believes that in the note of Figure 1 the “as-prepared Cu₂O nanocubes and Cu₂O-pyS” should be annotated as “(a)”, rather than “(d)”, as shown below.

Response: We apologize that we made some mistakes in the notes of figures. We have checked the entire manuscript carefully and made the corrections. The revised Figure 1 (**Figure R2-9**) is shown below:

Figure R2-9. FE of various products for the as-prepared Cu₂O nanocubes (a), the commercial Cu and c-Cu-pyS (b), commercial Cu₂O and c-Cu₂O-pyS (c), and commercial CuO and c-CuO-pyS (d). Error bars represent the standard deviation of 3 replicate measurements.

Reviewer #3 (Remarks to the Author):

In my opinion, the authors have addressed all concerns from the reviewer 1. Overall, the results presented are interesting, from both mechanistic and performance perspectives. I have some minor suggestions about the experimental details so other groups can reproduce the data.

Response: We sincerely appreciate the reviewer for the time and efforts spent in assessing our work. We have conducted a comprehensive study on the aspects raised by the reviewer. To fully address the concerns raised by the reviewer, we have provided detailed responses and supporting data as shown below.

1. The authors used a carbon gas diffusion layer that can operate for 100 hours without flooding problem. This is quite surprising. Details about this material should be provided.

Response: We thank the reviewer for the helpful suggestion. To ensure long-term stability, it is necessary to implement multiple steps for the integration of the catalyst with the support. The specific procedures are as follows:

- a. Firstly, applying a particle dispersion liquid (PTFE) onto carbon paper and then annealing it in air at 380 °C for half an hour. This step primarily aims to enhance the hydrophobicity of carbon paper without compromising gas diffusion.
- b. Spraying catalyst onto carbon paper and placing a nickel mesh on the side with PTFE brush layer to provide support. There will be complete physical contact between PTFE and carbon paper, while one side of Ni mesh will not meet electrolyte, thus minimizing HER.
- c. The final layer of carbon paper does not contain PTFE but serves as a conduit for gas conduction through direct physical contact. The three-layer structure consists of: first layer - carbon paper; second layer - nickel mesh; third layer - PTFE-coated carbon paper that requires slight compression during preparation.
- d. With a total thickness of around 3 mm, gas diffusion occurs evenly through the first layer of carbon paper and subsequently passes through both the catalyst layer and the second layer containing PTFE before reaching the reaction port.

2. The authors mentioned that the flow cell is 9 cm². Is this the area of the catalyst (electrode) or the size of the cell? Details about the flow cell configuration should be provided. From figure 2, it looks like there is no gas chamber or gas flow channel.

Response: We thank the reviewer for raising the question. In our experiment, the area of the electrode in the flow cell device is 9 cm². We sincerely apologize for any confusion caused to the reviewer. Figure 2 only shows a schematic diagram of the flow pool and does not provide intricate details. To add details, we have made modifications by including a comprehensive schematic of the device (**Figure R3-1**). Moreover, the flow pool reaction is further illustrated with a comprehensive diagram (**Figure R3-2**).

Figure R3-1. (a) CORR stability test of Cu₂O-pyS. Inset is the schematic illustration showing the flow cell. (b) Quasi *in-situ* time-dependent Cu LMM X-ray-excited Auger electron spectra of Cu₂O-pyS collected at -0.65 V vs. RHE. Quasi *in-situ* time-dependent high-resolution N 1s XPS spectra (c) and S 2p XPS spectra (d) collected at -0.65 V vs. RHE. (e) *In-situ* ATR-SEIRAS spectra from 1000 cm⁻¹ to 2300 cm⁻¹ recorded over Cu₂O-pyS in CO-saturated 0.1 M KOH solution at -0.65 V vs. RHE.

Figure R3-2. The schematic diagram of the electrochemical flow cell.

3. How did the authors collect liquid products (only in catholyte or in both catholyte and anolyte)?

Response: We thank the reviewer for raising the question. The CORR experiment was conducted in a three-channel flow cell, utilizing 0.1 M KOH as the electrolyte, as depicted in **Figure R3-3**. To prevent product crossover at the anode and cathode, a proton exchange membrane (Nafion® 117) was employed to separate the anode and cathode chambers. Consequently, acetate measurements were conducted only in the cathode chamber.

Figure R3-3. The photograph showing the CO reduction electrolyzer.

4. How did the author deposit catalyst ink onto the gas diffusion layer?

Response: We thank the reviewer for raising the question. To ensure precise control of the catalytic layer thickness and achieve good uniformity, we employed ultrasonic spraying method to prepare the electrode. This involves two steps: (1) Ultrasonic dispersion to prepare catalyst paste; (2) Atomization and ultrasonic spraying of the catalyst slurry onto a support body, which can be either a gas diffusion layer or proton exchange membrane.

REVIEWERS' COMMENTS

Reviewer #2 (Remarks to the Author):

The authors have adequately addressed all my comments.

Reviewer #3 (Remarks to the Author):

The authors have addressed all my suggestions.

Response to Reviewers

We are very grateful to the critical comments and constructive suggestions provided by the reviewers, which shall significantly help to improve the quality of our work. Our manuscript has been revised accordingly, and the changes are highlighted by yellow colour in the text. The following list our responses to the comments from the reviewers. All the original comments are given in blue colour (italic) and our responses in black colour.

Reviewer #2 (Remarks to the Author):

The authors have adequately addressed all my comments.

Response: We sincerely appreciate the reviewer for the time and efforts spent in evaluating our manuscript.

Reviewer #3 (Remarks to the Author):

The authors have addressed all my suggestions.

Response: We sincerely appreciate the reviewer for the time and efforts spent in evaluating our manuscript.